# Arbitrarily-Conditioned Multi-Functional Diffusion for Multi-Physics Emulation

**Da Long** [1]   **Zhitong Xu** [1]   **Guang Yang** [2]   **Akil Narayan** [2,3]   **Shandian Zhe** [1]

## Abstract

Modern physics simulation often involves multiple functions of interests, and traditional numerical approaches are known to be complex and computationally costly. While machine learning-based surrogate models can offer significant cost reductions, most focus on a single task, such as forward prediction, and typically lack uncertainty quantification — an essential component in many applications. To overcome these limitations, we propose Arbitrarily-Conditioned Multi-Functional Diffusion (ACM-FD), a versatile probabilistic surrogate model for multi-physics emulation. ACM-FD can perform a wide range of tasks within a single framework, including forward prediction, various inverse problems, and simulating data for entire systems or subsets of quantities conditioned on others. Specifically, we extend the standard Denoising Diffusion Probabilistic Model (DDPM) for multi-functional generation by modeling noise as Gaussian processes (GP). We propose a random-mask based, zero-regularized denoising loss to achieve flexible and robust conditional generation. We induce a Kronecker product structure in the GP covariance matrix, substantially reducing the computational cost and enabling efficient training and sampling. We demonstrate the effectiveness of ACM-FD across several fundamental multi-physics systems. The code is released at https://github.com/BayesianAIGroup/ACM-FD.

## 1 Introduction

Physical simulation plays a crucial role in numerous scientific and engineering applications. Traditional numerical approaches (Zienkiewicz et al., 1977; Mitchell & Griffiths, 1980), while offering strong theoretical guarantees, are often complex to implement and computationally expensive to run. In contrast, machine learning-based surrogate models — also known as emulators — are trained on simulation or measurement data and can significantly reduce computational costs, making them a promising alternative (Kennedy & O'Hagan, 2000; Razavi et al., 2012).

However, modern physical simulations often involve multiple functions of interest, such as initial and boundary conditions, solution or state functions, parameter functions, source functions, and more. Current machine learning-based surrogate models, such as neural operators (Li et al., 2020a; Lu et al., 2021; Kovachki et al., 2023), primarily focus on a single prediction task, for instance, forward prediction of the solution function. To perform other tasks, one typically needs to retrain a surrogate model from scratch. In addition, most existing methods do not support uncertainty quantification, which is important in practice. For example, confidence intervals are important to assess the reliability of emulation results, and in inverse problems, a posterior distribution is required since such problems are often ill-posed.

To address these limitations, we propose Arbitrarily-Conditioned Multi-Functional Diffusion (ACM-FD), a versatile probabilistic surrogate model for multi-physics emulation. Within a single framework, ACM-FD can handle a wide range of tasks, including forward prediction with different input functions, various inverse problems conditioned on different levels of information, simulating data for entire systems, and generating a subset of quantities of interest conditioned on others. As a generative model, ACM-FD produces predictive samples, naturally supporting uncertainty quantification across all contexts. The contributions of our work are summarized as follows.

- **Multi-Functional Diffusion Framework:** We propose a multi-functional diffusion framework based on the Denoising Diffusion Probabilistic Model (DDPM) (Ho et al., 2020). By modeling the noise as multiple Gaussian processes (GPs), we perform diffusion and denoising in functional spaces, enabling the generation of multiple functions required in multiphysics systems.

- **Innovative Denoising Loss:** We introduce a denois-

[1]Kahlert School of Computing, University of Utah [2]Department of Mathematics, University of Utah [3]Scientific Computing and Imaging Institute, University of Utah. Correspondence to: Shandian Zhe <zhe@cs.utah.edu>.

*Proceedings of the 42nd International Conference on Machine Learning*, Vancouver, Canada. PMLR 267, 2025. Copyright 2025 by the author(s).

ing loss that encapsulates all possible conditional parts within the system. During training, we repeatedly sample conditional components using random masks, training the denoising network not only to restore noise for the parts to be generated, but also to predict zero values for the conditioned components. This regularization stabilizes the network and prevents excessive perturbations in the conditioned components during generation. This way, ACM-FD can flexibly generate function values conditioned on any given set of functions or quantities, allowing it to tackle a wide range of tasks, including forward prediction, inverse inference, completion, and system simulation.

- **Efficient Training and Sampling**: To enable efficient training and sampling, we use a multiplicative kernel to induce a Kronecker product structure within the GP covariance matrix. By leveraging the properties of the Kronecker product and tensor algebra, we bypass the need to compute the full covariance matrix and its Cholesky decomposition, substantially reducing the training and sampling costs.
- **Experiments**: We evaluated ACM-FD on four fundamental multi-physics systems, with the number of involved functions ranging from three to seven. In twenty-four prediction tasks across these systems, ACM-FD consistently achieved top-tier performance compared to state-of-the-art neural operators specifically trained for each task. Furthermore, we evaluated ACM-FD in emulating all functions jointly and function completion. The data generated by ACM-FD not only closely adheres to the governing equations but also exhibits strong diversity. Its quality is comparable to that of a diffusion model trained exclusively for unconditional generation. In parallel, ACM-FD achieves substantially higher completion accuracy than popular inpainting and interpolation methods. ACM-FD also provides superior uncertainty calibration compared to alternative approaches. Finally, a series of ablation studies confirmed the effectiveness of the individual components of our method.

## 2 Preliminaries

The denoising diffusion probabilistic model (DDPM) (Ho et al., 2020) is one of the most successful generative models. Given a collection of data instances, such as images, DDPM aims to capture the complex underlying distribution of these instances and generate new samples from the same distribution. To achieve this, DDPM specifies a forward diffusion process that gradually transforms each data instance $\mathbf{x}_0$ into Gaussian white noise. The forward process is modeled as a Gauss-Markov chain,

$$q(\mathbf{x}_0, \ldots, \mathbf{x}_T) = q(\mathbf{x}_0) \prod_{t=1}^{T} q(\mathbf{x}_t | \mathbf{x}_{t-1}), \quad (1)$$

where $q(\mathbf{x}_0)$ represents the original data distribution, and each transition $q(\mathbf{x}_t | \mathbf{x}_{t-1}) = \mathcal{N}(\mathbf{x}_t | \sqrt{1 - \beta_t} \mathbf{x}_{t-1}, \beta_t \mathbf{I})$, with $\beta_t > 0$ as the noise level at step $t$. From this, it is straightforward to derive the relationship between the noisy instance $\mathbf{x}_t$ and the original instance $\mathbf{x}_0$,

$$\mathbf{x}_t = \sqrt{\widehat{\alpha}_t} \mathbf{x}_0 + \sqrt{1 - \widehat{\alpha}_t} \boldsymbol{\xi}_t, \quad \boldsymbol{\xi}_t \sim \mathcal{N}(\cdot | 0, \mathbf{I}), \quad (2)$$

where $\widehat{\alpha}_t = \prod_{j=1}^{t} \alpha_j$ and $\alpha_t = 1 - \beta_t$. As $t$ increases, $\widehat{\alpha}_t$ approaches zero, and $1 - \widehat{\alpha}_t$ approaches 1, indicating that $\mathbf{x}_t$ is gradually converging to a standard Gaussian random variable. When $t$ becomes sufficiently large, we can approximately view $\mathbf{x}_t$ as Gaussian white noise.

DDPM then learns to reverse this diffusion process to reconstruct the original instance $\mathbf{x}_0$ from the Gaussian white noise. Data generation, or sampling, is achieved by running this reversed process, which is often referred to as the denoising process. The reversed process is modeled as another Gauss-Markov chain, expressed as

$$p_{\Theta}(\mathbf{x}_T, \ldots, \mathbf{x}_0) = p(\mathbf{x}_T) \prod_{t=1}^{T} p_{\Theta}(\mathbf{x}_{t-1} | \mathbf{x}_t), \quad (3)$$

where $p(\mathbf{x}_T) = \mathcal{N}(\mathbf{x}_T | \mathbf{0}, \mathbf{I})$, and $\Theta$ denotes the model parameters. DDPM uses the variational learning framework (Wainwright et al., 2008), which leads to matching each transition $p_{\Theta}(\mathbf{x}_{t-1} | \mathbf{x}_t)$ to the reversed conditional distribution from the forward diffusion process,

$$q(\mathbf{x}_{t-1} | \mathbf{x}_t, \mathbf{x}_0)$$
$$= \mathcal{N}\left(\mathbf{x}_{t-1} | \frac{1}{\sqrt{1 - \beta_t}}\left(\mathbf{x}_t - \frac{\beta_t}{\sqrt{1 - \widehat{\alpha}_t}} \boldsymbol{\xi}_t\right), \widehat{\beta}_t \mathbf{I}\right), \quad (4)$$

where $\widehat{\beta}_t = \frac{1 - \widehat{\alpha}_{t-1}}{1 - \widehat{\alpha}_t} \beta_t$. Accordingly, DDPM employs a neural network $\Phi$ to approximate the noises $\boldsymbol{\xi}_t$; see (2). The training objective is to minimize the denoising loss, $\mathbb{E}_t \|\Phi_{\Theta}(\mathbf{x}_t, t) - \boldsymbol{\xi}_t\|^2$.

## 3 Method

We consider a multi-physics system involving a collection of $M$ functions. These functions may represent initial and boundary conditions, source terms, parameter functions, system states and others. Our goal is to capture the underlying complex relationships between these functions and perform a wide range of prediction and simulation tasks, such as forward prediction of the system state given the initial condition, inverse inference about the systems parameters given observed states, joint simulation of all the functions, or conditional simulation of one set of functions based on another.

Many of these tasks involve mapping between functions, making neural operators (Li et al., 2020a; Lu et al., 2021; Kovachki et al., 2023) — a recently developed class of surrogate models — an appealing approach. However, most

neural operators are trained for a single prediction task. To perform a different task, one would need to train another neural operator from scratch. These models generally lack uncertainty quantification, which is important for many applications. Furthermore, these models are unable to carry out data generation.

To address these issues, we propose ACM-FD, a new generative model that can serve as a powerful and versatile probabilistic emulator for multi-physics systems.

## 3.1 Multi-Functional Diffusion

We first extend DDPM to enable functional data generation. To this end, we generalize the diffusion process in (2) to the functional space. Given a function instance $f_0(\cdot)$, we gradually transform it into a noise function via

$$f_t(\cdot) = \sqrt{\widehat{\alpha}_t} f_0(\cdot) + \sqrt{1 - \widehat{\alpha}_t} \xi_t(\cdot), \tag{5}$$

where $f_t(\cdot)$ is the noisy version of $f_0$ at step $t$, and $\xi_t(\cdot)$ is the noise function used to corrupt $f_0$. To model the noise function, one can employ an expressive stochastic process. We choose to sample each noise function from a zero-mean Gaussian process (GP) (Rasmussen & Williams, 2006),

$$\xi_t \sim \mathcal{GP}(\cdot|0, \kappa(\mathbf{z}, \mathbf{z}')), \tag{6}$$

where $\kappa(\cdot, \cdot)$ is the covariance (kernel) function and $\mathbf{z}$ and $\mathbf{z}'$ denote the input locations of the function. When $t$ becomes sufficiently large (i.e., $\widehat{\alpha}_t \approx 0$), we can approximately view $f_t \sim \mathcal{GP}(0, \kappa(\mathbf{z}, \mathbf{z}'))$, meaning it effectively turns into a noise function.

The actual data is the function instance sampled at a finite set of locations, $\mathbf{f}_0 = f_0(\mathcal{Z}) = (f_0(\mathbf{z}_1), \ldots, f_0(\mathbf{z}_N))^\top$ where $\mathcal{Z} = \{\mathbf{z}_j | 1 \le j \le N\}$ are the sampling locations. Therefore, we only need to apply the diffusion process (5) to $\mathcal{Z}$ (with the function values at all the other locations marginalized out), which yields:

$$\mathbf{f}_t = \sqrt{\widehat{\alpha}_t} \mathbf{f}_0 + \sqrt{1 - \widehat{\alpha}_t} \boldsymbol{\xi}_t, \quad \boldsymbol{\xi}_t \sim \mathcal{N}(\cdot|0, \mathbf{K}), \tag{7}$$

where $\mathbf{f}_t = f_t(\mathcal{Z})$, $\boldsymbol{\xi}_t = \xi_t(\mathcal{Z})$, and $\mathbf{K} = \kappa(\mathcal{Z}, \mathcal{Z})$ is the covariance matrix at $\mathcal{Z}$.

Based on (7), we can derive the reversed conditional distribution,

$$q(\mathbf{f}_{t-1}|\mathbf{f}_t, \mathbf{f}_0) \tag{8}$$
$$= \mathcal{N}\left(\mathbf{f}_{t-1} | \frac{1}{\sqrt{1 - \beta_t}} \left(\mathbf{f}_t - \frac{\beta_t}{\sqrt{1 - \widehat{\alpha}_t}} \boldsymbol{\xi}_t\right), \widehat{\beta}_t \mathbf{K}\right).$$

Comparing to (4), the only difference is that the identity matrix is replaced by the covariance matrix $\mathbf{K}$ at the sampling locations $\mathcal{Z}$. We then use a similar strategy as in DDPM. We train a neural network $\Phi_\Theta$ such that $\Phi_\Theta(\mathbf{f}_t, t, \mathcal{Z}) \approx \boldsymbol{\xi}_t$,

and substitute the neural network's prediction for $\boldsymbol{\xi}_t$ in (8) to obtain $p_\Theta(\mathbf{f}_{t-1}|\mathbf{f}_t)$ for denoising and generation.

Next, we generalize to the multi-functional case. Given $M$ functions of interest, $\mathcal{F} = \{f_0^1(\cdot), \ldots, f_0^M(\cdot)\}$, we employ the same diffusion process as specified in (5) and (7) to transform each function into a noise function. Next, we use a single denoising neural network to jointly recover or sample all the functions, thereby capturing the complex and strong relationships among them. Specifically, we construct a network $\Phi_\Theta$ that takes the step $t$, all the noisy function values sampled at $t$, and the sampling locations as the input, to predict the corresponding noises,

$$\phi_\Theta(\mathbf{f}_t^1, \ldots, \mathbf{f}_t^M, t, \overline{\mathcal{Z}}) \approx (\boldsymbol{\xi}_t^1, \ldots, \boldsymbol{\xi}_t^M), \tag{9}$$

where each $\mathbf{f}_t^k = f_t^k(\mathcal{Z}_k)$, $\boldsymbol{\xi}_t^k = \xi_t^k(\mathcal{Z}_k)$, $f_t^k(\cdot)$ is the noisy version of $f_0^k(\cdot)$ at step $t$, $\xi_t^k(\cdot)$ is the corresponding noise function that corrupts $f_0^k$ to yield $f_t^k$, $\mathcal{Z}_k$ are the sampling locations of $f_0^k$, and $\overline{\mathcal{Z}} = \{\mathcal{Z}_k\}_{k=1}^M$.

## 3.2 Arbitrarily-Conditioned Denoising Loss

The design of our multi-functional denoising model (9) enables us to perform a wide range of conditional sampling and prediction tasks, beyond just functional data generation.

Specifically, let us denote an arbitrary set of conditioned function values as $\mathbf{F}^c = \{f_0^k(\mathcal{Z}_k^c) | k \in \mathcal{C}\}$, and the target function values to be generated as $\mathbf{F}^s = \{f_0^k(\mathcal{Z}_k^s) | k \in \mathcal{S}\}$. Here $\mathcal{C}$ and $\mathcal{S}$ denote the indices of the functions to be conditioned on and those to be generated, respectively. Note that $\mathcal{C}$ and $\mathcal{S}$ may overlap, provided that for any $k \in \mathcal{C} \cap \mathcal{S}$, the sampling locations do not, i.e., $\mathcal{Z}_k^c \cap \mathcal{Z}_k^s = \emptyset$. To generate $\mathbf{F}^s$ conditioned on $\mathbf{F}^c$, we fix $\mathbf{F}^c$ in the input to the network $\Phi_\Theta$ as a constant, and vary only the noisy state associated with $\mathbf{F}^s$, denoted as $\mathbf{F}_t^s$ (at step $t$). The input to $\Phi$ consists of $\mathbf{F}^c \cup \mathbf{F}_t^s \cup \{t, \overline{\mathcal{Z}}\}$. We use the predicted noise corresponding to $\mathbf{F}_t^s$ to sample $\mathbf{F}_{t-1}^s$, following the denoising process. This procedure is repeated until $t = 0$, resulting in a sample for $\mathbf{F}^s$. The sampling process is summarized in Algorithm 2.

By varying the choices of $\mathbf{F}^c$ and $\mathbf{F}^s$, our model can perform a wide variety of data generation and prediction tasks. When $\mathbf{F}^c = \emptyset$, the model jointly samples the $M$ target functions at the specified input locations. When $\mathbf{F}^c \ne \emptyset$ and $\mathcal{C} \cap \mathcal{S} = \emptyset$, the model generates one set of functions conditioned on the other, supporting various forward prediction and inverse inference tasks. For instance, by setting $\mathcal{C}$ to include initial conditions or current and/or past system states and source functions, and $\mathcal{S}$ to the future system states, the model performs forward prediction. Alternatively, if the observed system states are included in $\mathcal{C}$ and $\mathcal{S}$ corresponds to initial/boundary conditions, system parameters, or past states, the model carries out inverse inference. Even for the same type of tasks, the functions and the number of functions in $\mathcal{C}$, as well as the number of sampling locations

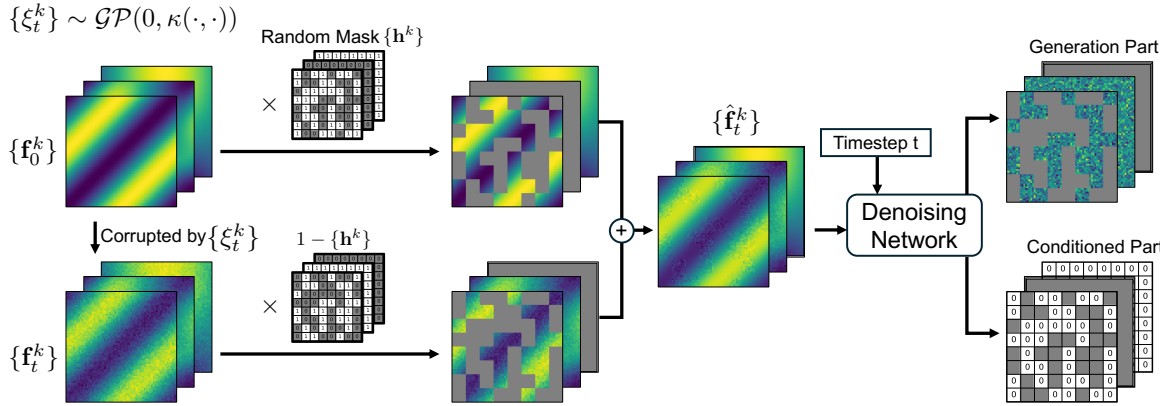

*Figure 1.* The illustration of ACM-FD. The grey shaded area represents zero values introduced by the random mask. The denoising network is trained to predict zeros for the conditioned part while recovering noises for the part to be generated (*i.e.*, unshaded noises).

in $\mathcal{Z}_k^c$, can be adjusted to accommodate different levels of available information for prediction. When $\mathcal{C} \cap \mathcal{S} \neq \emptyset$, the model performs not only conditional sampling of other functions but also completion of the observed functions at new locations. All tasks can be executed using a single model, $\Phi_\Theta$, *during inference*, following the same sampling process. One can generate as many predictive samples as needed, naturally supporting uncertainty quantification in arbitrary context. Thereby, our method offers a versatile probabilistic surrogate for multi-physics emulation.

However, to build a well-performing model, training $\Phi$ to predict all the noises as specified in (9) is far from sufficient, because it only handles unconditioned sampling. To train our model to handle all kinds of tasks, we introduce a random mask variable, $\mathcal{H} = \{\mathbf{h}^1, \dots, \mathbf{h}^M\}$. Each element in $\mathcal{H}$ is binary, indicating whether the corresponding function value is conditioned or to be sampled. If a function value is conditioned, then it should remain fixed as a constant input to $\Phi$, and correspondingly, the predicted noise for that value should be *zero*, since it is never corrupted by any noise. Accordingly, we introduce a new denoising loss,

$$\mathcal{L}(\Theta) \tag{10}$$
$$= \mathbb{E}_t \mathbb{E}_{p(\mathbf{H})} \left\| \Phi_\Theta(\widehat{\mathbf{f}}_t^1, \dots, \widehat{\mathbf{f}}_t^M, t, \overline{\mathcal{Z}}) - (\widehat{\boldsymbol{\xi}}_t^1, \dots, \widehat{\boldsymbol{\xi}}_t^M) \right\|^2,$$

where for each $1 \leq k \leq M$,

$$\widehat{\mathbf{f}}_t^k = \mathbf{f}_0^k \circ \mathbf{h}^k + \mathbf{f}_t^k \circ (\mathbf{1} - \mathbf{h}^k),$$
$$\widehat{\boldsymbol{\xi}}_t^k = \mathbf{0} \circ \mathbf{h}^k + \boldsymbol{\xi}_t^k \circ (\mathbf{1} - \mathbf{h}^k). \tag{11}$$

Here $\mathbf{f}_0^k$ is the original function instance sampled at the specified locations $\mathcal{Z}^k$, $\mathbf{f}_t^k = \sqrt{\widehat{\alpha}_t} \mathbf{f}_0^k + \sqrt{1 - \widehat{\alpha}_t} \boldsymbol{\xi}_t^k$ is the noisy version of $\mathbf{f}_0^k$ at step $t$, and $\circ$ denotes the element-wise product. From (11), we can see $\widehat{\boldsymbol{\xi}}_t^k$ is mixed with zeros for the conditioned part and the noises for the part to generate. The zeros in $\widehat{\boldsymbol{\xi}}_t^k$ regularizes the network $\Phi_\Theta$, preventing it from

producing excessively large perturbations in the conditioned part. This encourages the model to better adhere to the system's underlying mechanism (*e.g.*, governing equations) during generation. See Fig. 1 for an illustration.

The choice of $p(\mathcal{H})$ is flexible. We may randomly mask individual observed values within each function and/or randomly mask entire functions. In the absence of prior knowledge or specific preferences, a natural choice is to set the masking probability to 0.5, as it yields the maximum masking variance (*i.e.*, diversity). Our denoising loss (10) accommodates all possible conditional components for conditioned sampling as well as unconditioned sampling (*i.e.*, $\mathcal{H}$ consists of all zeros), allowing our model to be trained for all kinds of tasks. We use stochastic training, which involves randomly sampling the mask $\mathcal{H}$, applying it to the function and noise instances as specified in (11), and computing the stochastic gradient $\nabla_\Theta \left\| \Phi_\Theta(\widehat{\mathbf{f}}_t^1, \dots, \widehat{\mathbf{f}}_t^M, t, \overline{\mathcal{Z}}) - (\widehat{\boldsymbol{\xi}}_t^1, \dots, \widehat{\boldsymbol{\xi}}_t^M) \right\|^2$ to update the model parameters $\Theta$.

### 3.3 Efficient Training and Sampling

The training and generation with our model requires repeatedly sampling noise functions from GPs, as specified in (7). The sampling necessitates the Cholesky decomposition of the covariance matrix $\mathbf{K}$ at the input locations, which has a time complexity of $\mathcal{O}(N^3)$, where $N$ is the number of input locations. When $N$ is large, the computation becomes prohibitively expensive or even infeasible. However, large values of $N$ is not uncommon in realistic systems. For instance, generating a 2D function sample on a $128 \times 128$ mesh results in $N = 128 \times 128 = 16,384$.

To address this challenge, we use a multiplicative kernel to model the noise function. Given the input dimension $D$, we

construct the kernel with the following form,

$$\kappa(\mathbf{z}, \mathbf{z}') = \prod_{d=1}^{D} \kappa(z_j, z_j'). \tag{12}$$

Notably, the widely used Square Exponential (SE) kernel already exhibits this structure: $\kappa_{\mathrm{SE}}(\mathbf{z}, \mathbf{z}') = \exp(-\|\mathbf{z} - \mathbf{z}'\|^2 / l^2)$. We position the sampling locations for each target function $f_0^k(\cdot)$ on an $m_1 \times \ldots \times m_D$ mesh, denoted as $\mathcal{Z}_k = \boldsymbol{\gamma}_1 \times \ldots \times \boldsymbol{\gamma}_D$, where $\times$ denotes the Cartesian product, and each $\boldsymbol{\gamma}_d$ $(1 \leq d \leq D)$ comprises the $m_d$ input locations within dimension $d$. From the multiplicative kernel (12),

---

**Algorithm 1** Training($\mathcal{Z}_1, \ldots, \mathcal{Z}_M, p(\mathcal{H})$)

---

1: **repeat**
2:   Sample instances of the $M$ functions of interest over meshes $\{\mathcal{Z}_k\}_{k=1}^{M}$, denoted by $\{\mathbf{f}_0^1, \ldots, \mathbf{f}_0^M\}$.
3:   $t \sim \mathrm{Uniform}(1, \ldots, T)$
4:   Sample $M$ noise functions from GPs over $\{\mathcal{Z}_k\}_{k=1}^{M}$, denoted by $\{\boldsymbol{\xi}_t^1, \ldots, \boldsymbol{\xi}_t^M\}$, where each $\boldsymbol{\xi}_t^k$ corresponds to $\mathbf{f}_t^k$, using (12) and (13)
5:   Sample the mask $\mathcal{H} = \{\mathbf{h}^1, \ldots, \mathbf{h}^M\} \sim p(\mathcal{H})$
6:   Take gradient descent step on

$$\nabla_\Theta \left\| \Phi_\Theta(\widehat{\mathbf{f}}_t^1, \ldots, \widehat{\mathbf{f}}_t^M, t, \overline{\mathcal{Z}}) - (\widehat{\boldsymbol{\xi}}_t^1, \ldots, \widehat{\boldsymbol{\xi}}_t^M) \right\|^2,$$

   where $\overline{\mathcal{Z}} \triangleq \{\mathcal{Z}_k\}_{k=1}^{M}$, each $\widehat{\mathbf{f}}_t^k$ and $\widehat{\boldsymbol{\xi}}_t^k$ $(1 \leq k \leq M)$ are masked instances as defined in (11).
7: **until** Converged

---

**Algorithm 2** Generation (conditioned: $\mathbf{F}^c$, target: $\mathbf{F}^s$, target locations: $\overline{\mathcal{Z}}^s$, all locations: $\overline{\mathcal{Z}} = \{\mathcal{Z}_k\}$)

---

1: Sample noise functions from GPs over $\overline{\mathcal{Z}}$ using (12) and (13), denoted as $\overline{\boldsymbol{\xi}}$
2: $\mathbf{F}_T^s \leftarrow$ Subset of $\overline{\boldsymbol{\xi}}$ at $\overline{\mathcal{Z}}^s$
3: **for** $t = T, \ldots, 1$ **do**
4:   $\overline{\boldsymbol{\epsilon}} \leftarrow \mathbf{0}$
5:   **if** $t > 1$ **then**
6:     Re-sample $\overline{\boldsymbol{\xi}}$ following STEP 1
7:     $\overline{\boldsymbol{\epsilon}} \leftarrow$ Subset of $\overline{\boldsymbol{\xi}}$ at $\overline{\mathcal{Z}}^s$
8:   **end if**
9:   $\overline{\boldsymbol{\xi}}_t \leftarrow \Phi_\Theta(\mathbf{F}^c \cup \mathbf{F}_t^s, t, \overline{\mathcal{Z}})$
10:   $\overline{\boldsymbol{\xi}}_t^s \leftarrow$ Subset of $\overline{\boldsymbol{\xi}}_t$ at $\overline{\mathcal{Z}}^s$
11:   Generate sample

$$\mathbf{F}_{t-1}^s = \frac{1}{\sqrt{1 - \beta_t}} \left( \mathbf{F}_t^s - \frac{\beta_t}{\sqrt{1 - \widehat{\alpha}_t}} \overline{\boldsymbol{\xi}}_t^s \right) + \sqrt{\widehat{\beta}_t} \overline{\boldsymbol{\epsilon}}$$

12: **end for**
13: **return** $\mathbf{F}_0^s$

---

we can induce a Kronecker product in the covariance matrix,

namely $\mathbf{K} \triangleq \kappa(\mathcal{Z}_k, \mathcal{Z}_k) = \mathbf{K}_1 \otimes \ldots \otimes \mathbf{K}_D$ where each $\mathbf{K}_d = \kappa(\boldsymbol{\gamma}_d, \boldsymbol{\gamma}_d)$. We then perform the Cholesky decomposition on each local kernel matrix, yielding $\mathbf{K}_d = \mathbf{L}_d \mathbf{L}_d^\top$. Utilizing the properties of Kronecker products, we can derive that $\mathbf{K}^{-1} = (\mathbf{L}_1^{-1})^\top \mathbf{L}_1^{-1} \otimes \ldots \otimes (\mathbf{L}_D^{-1})^\top \mathbf{L}_D^{-1} = \mathbf{A}^\top \mathbf{A}$ where $\mathbf{A} = \mathbf{L}_1^{-1} \otimes \ldots \otimes \mathbf{L}_D^{-1}$. Consequently, to generate a sample of $\xi_t$ on the mesh, we can first sample a standard Gaussian variable $\boldsymbol{\eta} \sim \mathcal{N}(\mathbf{0}, \mathbf{I})$, and then obtain the sample as $\mathrm{vec}(\boldsymbol{\xi}_t) = \mathbf{A}^\top \boldsymbol{\eta}$ where $\mathrm{vec}(\cdot)$ denotes the vectorization. Note that $\boldsymbol{\xi}_t$ is a $m_1 \times \ldots \times m_D$ tensor. However, directly computing the Kronecker product in $\mathbf{A}$ and then evaluating $\mathbf{A}^\top \boldsymbol{\eta}$ can be computationally expensive, with a time complexity of $\mathcal{O}(\overline{m}^2)$, where $\overline{m} = \prod_d m_d$ represents the total number of the mesh points. To reduce the cost, we use tensor algebra (Kolda, 2006) to compute the Tucker product instead, which gives the same result without explicitly computing the Kronecker product,

$$\boldsymbol{\xi}_t^k = \Pi \times_1 \mathbf{L}_1^{-1} \times_2 \ldots \times_D \mathbf{L}_D^{-1}, \tag{13}$$

where we reshape $\boldsymbol{\eta}$ into an $m_1 \times \ldots \times m_D$ tensor $\Pi$, and $\times_k$ is the tensor-matrix product along mode $k$. The time complexity of this operation becomes $\mathcal{O}(\overline{m}(\sum_d m_d))$. Overall, this approach avoids computing the full covariance matrix $\mathbf{K}$ for all the mesh points along with its Cholesky decomposition, which would require $\mathcal{O}(\overline{m}^3)$ time and $\mathcal{O}(\overline{m}^2)$ space complexity. Instead we only compute the local kernel matrix for each input dimension. For example, when using a $128 \times 128$ mesh, the full covariance matrix would be $128^2 \times 128^2$, which is highly expensive to compute. In contrast, our approach requires the computation of only two local kernel matrices, each of size $128 \times 128$. Our approach further uses the Tucker product to compute the noise function sample (see (13)), avoiding explicitly computing the expensive Kronecker product. As a result, the overall time and space complexity is reduced to $\mathcal{O}(\sum_d m_d^3 + \overline{m}(\sum_d m_d))$ and $\mathcal{O}(\sum_d m_d^2 + \overline{m})$, respectively. This enables efficient training and generation of our model. Finally, we summarize our method in Algorithm 1 and 2.

## 4 Related Work

Generative modeling is a fundamental topic in machine learning. The DDPM (Ho et al., 2020) is recent a breakthrough, and numerous subsequent works have expanded upon this direction, further advancing the field, such as score-based diffusion via SDEs (Song et al., 2021b), denoising diffusion implicit models (DDIM) (Song et al., 2021a), and flow-matching (Lipman et al., 2022; Klein et al., 2023). Recent works have explored diffusion models for inverse problems, such as (Chung et al., 2023; Wang et al., 2023a; Song et al., 2023). However, these approaches mainly train an unconditional diffusion model, and then perform conditional sampling at the inference stage.

A few recent works have explored diffusion models for

function generation, such as (Kerrigan et al., 2023) based on DDPM, (Lim et al., 2023; Franzese et al., 2024) leveraging score- or SDE-based diffusion, (Zhang & Wonka, 2024) using DDIM, and (Kerrigan et al., 2024) with flow matching. Our approach, which incorporates a GP noise function into the DDPM framework, is conceptually similar to (Kerrigan et al., 2023), where a GP is treated as a special case of a Gaussian random element in their formulation. However, the key distinction lies in the scope of our work. Unlike prior methods focused on single-function generation, our approach is designed for multi-functional scenarios, simultaneously addressing diverse single- and multi-functional generation tasks conditioned on varying quantities and/or functions during inference. To achieve this, we propose an arbitrarily conditioned multi-functional diffusion framework that integrates a random masking strategy, a zero-regularized denoising loss, and an efficient training and sampling method. Our work is related to (Gloeckler et al., 2024), which leverages score-based diffusion for simulation and also uses a random masking strategy to fulfill flexible conditional sampling. But (Gloeckler et al., 2024) does not enforce zero regularization on the score model's predictions for the conditioned part. In contrast, our method incorporates zero regularization in the loss function to prevent the denoising network from producing excessive artificial noise to the conditioned part, thereby enhancing stability. A detailed comparison is provided in Section 5. Another related work is (Wang et al., 2023b), which introduces a score-based diffusion model for multi-fidelity simulation. However, the approach is not formulated as function generation and is limited to a single prediction task.

Neural operator (NOs) aim to learn function-to-function mappings from data, primarily for estimating PDE operators from simulation data and performing forward predictions of PDE solutions given new inputs (*e.g.,* parameters, initial and/or boundary conditions, source terms). Important works include Fourier Neural Operators (FNO) (Li et al., 2022) and Deep Operator Net (DONet) (Lu et al., 2021), which have demonstrated promising prediction accuracy across various benchmark PDEs (Lu et al., 2022). Many NO models have been developed based on FNO and DONet, such as (Gupta et al., 2021; Wen et al., 2022; Lu et al., 2022; Tran et al., 2023). Li et al. (2024) developed an active learning method to query multi-resolution data for enhancing FNO training while reducing the data cost. Recent advances in kernel operator learning have been made by Long et al. (2022), Batlle et al. (2023), and Lowery et al. (2024).

## 5  Experiments

For evaluation, we considered four fundamental multi-physics systems:

- **Darcy Flow (D-F)**: a single-phase 2D Darcy flow sys-

tem that involves three functions: the permeability field $a$, the flow pressure $u$, and the source term $f$.
- **Convection Diffusion (C-D)**: a 1D convection-diffusion system that involves three spatial-temporal functions: the scalar field of the quantity of interest $u$, the velocity field $v$, and the source function $s$.
- **Diffusion Reaction (D-R)**: a 2D diffusion-reaction system, for which we are interested in four spatial functions: $f_1$ and $f_2$, which represent the activator and inhibitor functions at time $t = 2.5$, and $u_1$ and $u_2$, denoting the activator and inhibitor at $t = 5.0$.
- **Torus Fluid (T-F)**: a viscous, incompressible fluid in the unit torus, for which we are interested in seven functions, including the source function $f$, the initial vorticity field $w_0$, and the vortocity fields $w_1, \ldots, w_5$ at five different time steps $t = 2, 4, 6, 8, 10$.

The details about these systems, along with the preparation of training and test data, are provided in Appendix Section A.

### 5.1  Predictive Performance Across Various Tasks

We first examined performance of ACM-FD across a variety of prediction tasks, each predicting one function using another set of functions. These tasks address a wide range of forward prediction and inverse inference problems. For example, in the case of *Darcy flow*, predicting the pressure field $u$ given the permeability field $a$ and the forcing term $f$ is a forward prediction task, while predicting $a$ from $u$ and $f$, or predicting $f$ from $u$ and $a$, typically represents inverse problems. Due to the versatility of ACM-FD, a single trained model can be used to perform all these tasks.

We compared with the following popular and state-of-the-art operator learning methods: (1) Fourier Neural Operator (FNO) (Li et al., 2020b), which performs channel lifting, followed by a series of Fourier layers that execute linear functional transformations using the fast Fourier transform and then apply nonlinear transformations, ultimately culminating in a projection layer that produces the prediction. (2) Deep Operator Net (DON) (Lu et al., 2021), which employs a branch-net and a trunk-net to extract representations of the input functions and querying locations, producing the prediction by the dot product between the two representations. (3) PODDON (Lu et al., 2022), a variant of DON where the trunk-net is replaced by the POD (PCA) bases extracted from the training data. (4) GNOT (Hao et al., 2023), a transformer-based neural operator that uses cross-attention layers to aggregate multiple input functions' information for prediction. We utilized the original implementation of each competing method. In addition, we compared with (5) Simformer (Gloeckler et al., 2024), an attention based emulator which also uses a random masking strategy to fulfill flexible conditional generation. However, Simformer does not fit

*Table 1.* Relative $L_2$ error for various prediction tasks. The results were averaged over five runs.

| Dataset | Task(s) | ACM-FD | FNO | GNOT | DON | Simformer |
|---|---|---|---|---|---|---|
| D-F | $f, u$ to $a$ | **1.32e-02 (2.18e-04)** | 1.88e-02 (1.66e-04) | 1.35e-01 (6.57e-05) | 2.38e-02 (3.45e-04) | 1.18e-01 (3.00e-03) |
| | $a, u$ to $f$ | **1.59e-02 (1.59e-04)** | 2.37e-02 (1.87e-04) | 1.00e+00 (0.00e+00) | 3.76e-02 (7.75e-04) | 4.11e-02 (2.87e-03) |
| | $a, f$ to $u$ | **1.75e-02 (4.16e-04)** | 6.29e-02 (4.18e-04) | 6.09e-01 (2.40e-01) | 6.05e-02 (7.17e-04) | 4.04e-02 (5.17e-03) |
| | $u$ to $a$ | **3.91e-02 (7.08e-04)** | 5.57e-02 (4.16e-04) | 1.35e-01 (1.99e-04) | 5.08e-02 (5.91e-04) | 1.44e-01 (4.23e-03) |
| | $u$ to $f$ | **3.98e-02 (6.45e-04)** | 5.50e-02 (5.47e-04) | 9.99e-01 (7.48e-04) | 6.46e-02 (1.13e-04) | 1.06e-01 (3.98e-03) |
| C-D | $s, u$ to $v$ | **2.17e-02 (4.53e-04)** | 4.50e-02 (3.89e-04) | 3.26e-02 (3.41e-03) | 3.64e-02 (5.07e-04) | 3.96e-01 (4.79e-02) |
| | $v, u$ to $s$ | **5.45e-02 (1.40e-03)** | 7.93e-02 (8.48e-04) | 1.22e-01 (1.91e-03) | 7.04e-02 (7.53e-04) | 5.76e-02 (7.10e-02) |
| | $v, s$ to $u$ | 1.60e-02 (2.15e-04) | 7.26e-02 (2.16e-04) | **5.80e-03 (1.51e-04)** | 7.86e-02 (7.42e-04) | 1.03e-01 (1.95e-02) |
| | $u$ to $v$ | **2.66e-02 (3.08e-04)** | 5.90e-02 (8.22e-04) | 6.69e-02 (3.66e-03) | 4.55e-02 (6.09e-04) | 5.108e-01 (7.56e-02) |
| | $u$ to $s$ | **6.06e-02 (2.54e-04)** | 1.16e-01 (5.63e-04) | 1.85e-01 (2.84e-03) | 9.65e-02 (5.52e-04) | 9.21e-01 (1.00e-01) |
| D-R | $f_1, u_1$ to $f_2$ | 1.44e-02 (8.96e-04) | **1.07e-02 (1.92e-04)** | 4.53e-01 (4.34e-02) | 2.93e-01 (1.29e-03) | 3.39e-02 (2.97e-03) |
| | $f_1, u_1$ to $u_2$ | **1.59e-02 (3.68e-04)** | 2.02e-02 (2.42e-04) | 3.91e-01 (1.86e-02) | 2.03e-01 (2.22e-03) | 3.67e-02 (2.36e-03) |
| | $f_2, u_2$ to $f_1$ | **4.10e-02 (8.93e-04)** | 5.52e-02 (3.01e-03) | 6.53e-01 (2.04e-02) | 4.24e-01 (9.26e-04) | 1.21e-01 (3.11e-03) |
| | $f_2, u_2$ to $u_1$ | **5.86e-02 (3.43e-04)** | 7.82e-02 (1.29e-04) | 4.88e-01 (2.92e-02) | 2.98e-01 (2.61e-03) | 1.01e-01 (2.70e-03) |
| T-F | $w_0, w_5$ to $w_1$ | 2.73e-02 (4.78e-03) | **1.28e-02 (2.38e-04)** | 2.40e-02 (8.74e-04) | 6.32e-02 (2.72e-04) | 6.14e-02 (2.44e-03) |
| | $w_0, w_5$ to $w_2$ | 2.43e-02 (1.60e-03) | **2.08e-02 (9.80e-05)** | 4.00e-02 (5.92e-04) | 7.69e-02 (4.41e-04) | 6.99e-02 (2.18e-03) |
| | $w_0, w_5$ to $w_3$ | 2.43e-02 (3.17e-03) | **2.33e-02 (1.83e-04)** | 4.74e-02 (1.23e-03) | 7.34e-02 (2.88e-04) | 8.34e-02 (2.60e-03) |
| | $w_0, w_5$ to $w_4$ | 1.68e-02 (1.81e-03) | **1.41e-02 (1.17e-04)** | 3.95e-02 (6.73e-04) | 5.57e-02 (1.73e-04) | 9.75e-02 (3.93e-03) |
| | $w_0, w_5$ to $f$ | **1.63e-02 (1.49e-03)** | 1.79e-02 (3.04e-04) | 5.91e-02 (4.01e-03) | 4.77e-02 (5.56e-04) | 1.14e-01 (4.00e-03) |
| | $w_0, f$ to $w_1$ | 3.10e-02 (4.08e-03) | **9.68e-03 (3.22e-04)** | 2.09e-02 (3.62e-04) | 6.08e-02 (3.14e-04) | 6.06e-02 (2.03e-03) |
| | $w_0, f$ to $w_2$ | 3.28e-02 (4.79e-03) | **1.70e-02 (3.51e-04)** | 4.15e-02 (8.21e-04) | 7.73e-02 (6.18e-04) | 6.18e-02 (1.02e-03) |
| | $w_0, f$ to $w_3$ | 3.49e-02 (2.38e-03) | **2.38e-02 (8.37e-05)** | 5.61e-02 (8.23e-04) | 8.82e-02 (4.45e-04) | 5.67e-02 (1.83e-03) |
| | $w_0, f$ to $w_4$ | 3.34e-02 (3.87e-03) | **3.10e-02 (1.26e-04)** | 6.97e-02 (1.62e-03) | 1.02e-01 (7.28e-04) | 4.10e-02 (1.98e-03) |
| | $w_0, f$ to $w_5$ | **3.26e-02 (2.13e-03)** | 3.81e-02 (2.01e-04) | 8.35e-02 (7.33e-04) | 1.21e-01 (8.20e-04) | 1.18e-01 (4.15e-03) |

zero values for the conditioned part as in our model; see Equations (10) and (11). The original Simfomer was developed with score-based diffusion (Song et al., 2021b). For a fair comparison, we adapted Simformer to the DDPM framework, and reimplemented it using PyTorch. Our method, ACM-FD, was implemented with PyTorch as well. For GP noise function sampling, we used the Square-Exponential (SE) kernel. In all the experiments, we used FNO to construct our denoising network $\Phi_\Theta$. Thereby, our comparison with FNO can exclude the factors from the architecture design. We combined two types of random masking: one that masks individual observed values within each function, and the other that masks entire functions, both with a masking probability of 0.5. The corresponding denoising losses for these two masking strategies (see (10)) are summed to form the final loss used for training the model.

We utilized 1,000 instances for training, 100 instances for validation, and 200 instances for testing. Hyperparameter tuning was performed using the validation set. The details are provided in Appendix Section B. Following the evaluation procedure in (Lu et al., 2022), for each method, we selected the optimal hyperparameters, and then conducted stochastic training for five times, reporting the average relative $L_2$ test error along with the standard deviation. The results are presented in Table 1. Due to the space limit, the comparison results with PODDON are provided in Appendix Table 5.

As we can see, ACM-FD consistently achieves top-tier pre-

dictive performance across all the tasks. In Darcy Flow (D-F), Convection Diffusion (C-D) and Diffusion Reaction (D-R), ACM-FD outperforms FNO in all tasks except for the $f_1, u_1$ to $f_2$ mapping. In the Torus Fluid (T-F) system, while ACM-FD ranks second, its performance remains close to that of FNO. Furthermore, the performance of ACM-FD is consistently better than Simformer and other neural operators, including GNOT, DON and PODDON (see Table 5 in the Appendix). Despite being specifically optimized for each task, those neural operators are constantly worse than our method, underscoring the advantage of training a single, versatile model. The observed improvement of Simformer over GNOT in many cases (both are based on transformer architectures) also implied the effectiveness of this strategy.

## 5.2 Generation Performance

We then evaluated the generation performance of ACM-FD by using it to generate instances of all relevant functions for the Darcy Flow, Convection Diffusion and Torus Fluid systems. We compared ACM-FD with two methods. The first one is multi-function diffusion (MFD) *solely* for generating all the functions. We used the same architecture as ACM-FD but trained the model unconditionally, in accordance with the standard DDPM framework (see (9)). The second method is $\beta$-VAE (Higgins et al., 2022), a widely used Variational Auto-Encoder (VAE), where $\beta$ controls the regularization strength from the prior. The encoder network was built using a series of convolutional layers, while the

decoder network was constructed as the transpose of these layers. For training and hyperparameter tuning of $\beta$-VAE, we used the same training and validation sets as those used by ACM-FD in Section 5.1. The details are provided in Appendix Section B.

We used two metrics to evaluate the quality of the generated data. The first is the *equation error*, which measures *how well the generated data adheres to the governing equations of the system*. To calculate this error, for each group of sampled source function, parameter function, and/or initial conditions, we ran the numerical solver — the same one used to generate the training data — to solve the governing equation(s), and then computed the relative $L_2$ error between the generated solution and the numerical solution. The second metric is the Mean-Relative-Pairwise-Distance (MRPD), which measures the diversity of the generated data (Yuan & Kitani, 2020; Barquero et al., 2023; Tian et al., 2024). It is worth noting that since there is no large pre-trained inception network tailored to encompass all types of functions across numerous multi-physics systems, the popular FID score (Heusel et al., 2017) is not applicable to our evaluation.

For each system, we generated 1,000 sets of the relevant functions. The average equation error and MRPD are presented in Table 2. As shown, the functions generated by ACM-FD and MFD consistently exhibit relatively small equation errors, indicating strong alignment with the physical laws governing each system. In contrast, the equation errors for functions generated by $\beta$-VAE are significantly larger, often by an order of magnitude. Furthermore, the data generated by ACM-FD and MFD demonstrates a higher MRPD compared to $\beta$-VAE, indicating better diversity among the generated function instances. For the functions generated by ACM-FD and MFD, both the equation error and MRPD are comparable. Specifically, in Darcy Flow (D-F) and Convection Diffusion (C-D), ACM-FD achieves smaller equation errors and higher MRPD, while in Torus Fluid (T-F), the metrics for ACM-FD are slightly worse. These results suggest that ACM-FD, despite being trained for a variety of prediction and conditional generation tasks, still achieves comparable unconditional generation performance to the diffusion model trained purely for un-conditional generation. Overall, these findings indicate that ACM-FD is capable of producing multi-physics data that is not only reliable but also diverse. In Appendix Section C, we provide examples of the functions generated by each method, along with detailed discussion and analysis.

### 5.3 Completion Performance

Third, we evaluated ACM-FD on the task of completing functions in unobserved domains. Specifically, we tested ACM-FD on Darcy Flow (D-F) and Convection Diffusion

*Table 2.* Equation relative $L_2$ error and diversity of generated data for the whole system. MRPD is short for Mean Relative Pairwise Distance.

| System | Task(s) | ACM-FD | MFD | $\beta$-VAE |
|---|---|---|---|---|
| D-F | Equation Error | **0.0576** | 0.0584 | 0.265 |
| | MRPD | **1.15** | 0.980 | 0.932 |
| C-D | Equation Error | **0.114** | 0.127 | 0.282 |
| | MRPD | **1.00** | 0.971 | 0.879 |
| T-F | Equation Error | 0.0273 | **0.0234** | 0.737 |
| | MRPD | 0.8042 | **0.9537** | 0.524 |

*Table 3.* Relative $L_2$ error for function completion. Each function is sampled in one half of the domain, while the other half is completed using different methods.

| Dataset | Task(s) | ACM-FD | MFD-Inpaint | Interp |
|---|---|---|---|---|
| D-F | $a$ | **1.21e-02** | 7.94e-02 | 1.04e-01 |
| | $f$ | **1.23e-02** | 6.41e-02 | 6.98e-01 |
| | $u$ | **1.09e-02** | 2.71e-02 | 8.07e-01 |
| C-D | $v$ | **1.87e-02** | 4.71e-01 | 8.30e-01 |
| | $s$ | **3.39e-02** | 3.22e-01 | 6.49e-01 |
| | $u$ | **1.45e-02** | 3.47e-02 | 8.97e-01 |

(C-D) systems. For each function, we provided samples from half of the domain and used ACM-FD to predict the function values in the other half. To benchmark performance, we adapted our multi-functional diffusion (MFD) framework to the DDPM inpainting approach proposed in (Lugmayr et al., 2022), which we refer to as MFD-Inpaint. This method trains MFD unconditionally as in Section 5.2. During the completion process, the observed function values are perturbed using the forward process, and these perturbed values, along with noise for the unobserved function values, are passed into the denoising network. Additionally, we compared ACM-FD with the interpolation strategy, which is commonly used in scientific and engineering domains. We employed `scipy.interpolate.griddata` for interpolation. We denote this method as Interp. The relative $L_2$ error for each method is presented in Table 3. As shown, ACM-FD consistently achieves much lower errors compared to both MFD-Inpaint and Interp. The superior performance of ACM-FD over MFD-Inpaint might be attributed to its explicitly conditional training strategy, which allows ACM-FD to leverage conditioned information more effectively during completion. It is also noteworthy that both ACM-FD and MFD-Inpaint substantially outperform classical inter-polation, demonstrating that diffusion-based modeling is far more effective for inferring unknown function values. Completion examples for each method are visualized in Appendix Fig. 6.

Table 4. Empirical Coverage Probability (ECP) under confidence levels $\alpha \in \{0.9, 0.95, 0.99\}$. Best results per task are in boldface.

| Dataset | Task | Method | 0.9 | 0.95 | 0.99 |
|---|---|---|---|---|---|
| C-D | $s, u$ to $v$ | ACM-FD | **0.833** | **0.880** | **0.921** |
| | | Simformer | 0.736 | 0.814 | 0.871 |
| | $v, u$ to $s$ | ACM-FD | **0.766** | **0.842** | **0.913** |
| | | Simformer | 0.683 | 0.767 | 0.879 |
| | $v, s$ to $u$ | ACM-FD | **0.939** | **0.968** | **0.990** |
| | | Simformer | 0.695 | 0.771 | 0.858 |
| | $u$ to $v$ | ACM-FD | **0.821** | **0.870** | **0.922** |
| | | Simformer | 0.775 | 0.850 | 0.912 |
| | $u$ to $s$ | ACM-FD | **0.920** | **0.949** | **0.972** |
| | | Simformer | 0.716 | 0.773 | 0.823 |
| D-F | $a, u$ to $f$ | ACM-FD | **0.947** | **0.974** | **0.991** |
| | | Simformer | 0.829 | 0.895 | 0.950 |
| | $a, f$ to $u$ | ACM-FD | **0.985** | **0.994** | **0.998** |
| | | Simformer | 0.922 | 0.955 | **0.998** |
| | $u$ to $f$ | ACM-FD | 0.867 | 0.909 | 0.952 |
| | | Simformer | **0.918** | **0.953** | **0.980** |

## 5.4 Uncertainty Quantification

Fourth, we assessed the quality of uncertainty calibration provided by ACM-FD. To this end, we examined the empirical coverage probability (Dodge, 2003): ECP = $\frac{1}{N} \sum_{i=1}^{N} \mathbb{I}(y_i \in C_\alpha)$, where $y_i$ is the ground-truth function value, and $C_\alpha$ is the $\alpha$ confidence interval derived from 100 predictive samples generated by our method. We varied $\alpha$ from $\{90\%, 95\%, 99\%\}$, and examined our method in eight prediction tasks across the Convection-Diffusion (C-D) and Darcy Flow (D-F) systems. We compared against Simformer. Note that all the other competing methods are deterministic and therefore unable to perform uncertainty quantification. As shown in Table 4, in most cases our method achieves coverage much closer to $\alpha$, showing superior quality in the estimated confidence intervals.

Additionally, we visualized prediction examples along with their uncertainties, measured by predictive standard deviation (std). As shown in Figure 2, more accurate predictions tend to have lower std, i.e., low uncertainty, while regions with larger prediction errors correspond to higher std — providing further qualitative evidence that our uncertainty estimates are well-aligned with prediction quality.

## 5.5 Ablation Studies

Finally, we conducted ablation studies to assess the contribution of the key components in ACM-FD, including *random masking strategy*, *masking probability $p$*, and *Kronecker product-based computation*. The results and analysis, as presented in Appendix Section D, confirm the effectiveness of each component.

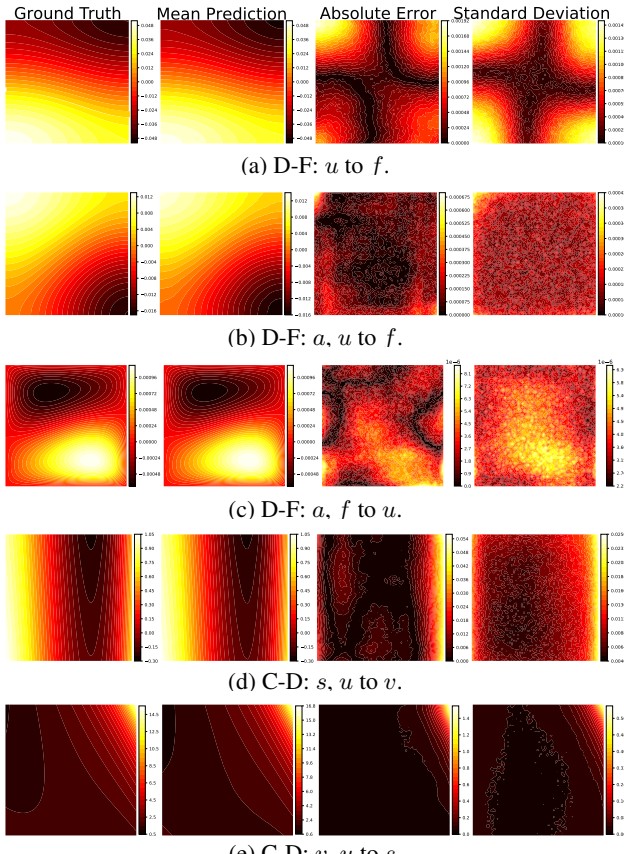

(a) D-F: $u$ to $f$.

(b) D-F: $a, u$ to $f$.

(c) D-F: $a, f$ to $u$.

(d) C-D: $s, u$ to $v$.

(e) C-D: $v, u$ to $s$.

Figure 2. Visualizations of ACM-FD predictions.

## 6 Conclusion

We have presented ACM-FD, a novel probabilistic generative model designed for multi-physics emulation. ACM-FD can perform arbitrarily-conditioned multi-function sampling and has demonstrated potential in several classical systems. In future work, we plan to extend our investigations into more complex multi-physics systems and explore alternative architecture designs to further extend its capabilities.

## Acknowledgements

SZ, DL and ZX acknowledge the support of MURI AFOSR grant FA9550-20-1-0358, NSF CAREER Award IIS-2046295, NSF OAC-2311685, and Margolis Foundation. AN acknowledges the support of the grant AFOSR FA9550-23-1-0749. This work has also leveraged NCSA Delta GPU cluster from NCF Access program under Project CIS240255.

## Impact Statement

This paper presents work whose goal is to advance machine learning for multi-physics emulation. There are many po-

tential societal consequences of our work, none of which we feel must be specifically highlighted here.

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

# Appendix

## A    Mulit-Physics Systems Details

### A.1    Darcy Flow

We considered a single-phase 2D Darcy flow system, which is governed by the following PDE,

$$-\nabla \cdot (a(\mathbf{x})\nabla u(\mathbf{x})) = f(\mathbf{x}) \quad \mathbf{x} \in (0,1)^2$$
$$u(\mathbf{x}) = 0, \quad \mathbf{x} \in \partial(0,1)^2, \tag{14}$$

where $a(\mathbf{x})$ is the permeability field, $u(\mathbf{x})$ is the fluid pressure, and $f(\mathbf{x})$ is the external source, representing sources or sinks of the fluid. To obtain one instance of the triplet $(a, f, u)$, we sampled $f$ from a Gauss random field: $f \sim \mathcal{N}(0, (-\Delta + 25)^{-\frac{15}{2}})$, and $a$ from the exponential of a Gaussian random field: $a = \exp(g)$ where $g \sim \mathcal{N}(0, (-\Delta + 16)^{-2})$. We then ran a second-order finite-difference solver to obtain $u$ at a $256 \times 256$ mesh. We extracted each function instance from a $64 \times 64$ sub-mesh.

### A.2    Convection Diffusion

We considered a 1D convection-diffusion system, governed by the following PDE,

$$\frac{\partial u(x,t)}{\partial t} + \nabla \cdot (v(x,t)u(x,t)) = D\nabla^2 u(x,t) + s(x,t), \tag{15}$$

where $(x,t) \in [-1,1] \times [0,1]$, $u(x,0) = 0$, $D$ is the diffusion coefficient and was set to 0.01, $v(x,t)$ is the convection velocity, describing how fast the substance is transported due to the flow, $s(x,t)$ is the source term, representing the external force, and $u(x,t)$ is the quantity of interest, such as the temperature, concentration and density. We employed a parametric form for $v$ and $a$: $v(x,t) = \sum_{n=1}^{3} a_n x^n + a_4 t$, and $s(x,t) = \alpha \exp(-\beta(x+t)^2) + \gamma \cos(\eta \cdot \pi(x - 0.1t))$, where all $a_n$, $\alpha, \beta, \gamma, \eta$ are sampled from Uniform$(-1,1)$. We used the MATLAB PDE solver pdepe[1] to obtain the solution for $u$. The spatial-temporal domain was discretized into a $512 \times 256$ mesh. We extracted the function instances of interest from a $64 \times 64$ equally-spaced sub-mesh.

### A.3    Diffusion Reaction

We used the 2D diffusion-reaction dataset provided from PDEBench (Takamoto et al., 2022). The dataset was simulated from a 2D diffusion-reaction system, which involves an activator function $v_1(t,x,y)$ and an inhibitor function $v_2(t,x,y)$. These two functions are non-linearly coupled. The governing equation is given as follows.

$$\frac{\partial v_1}{\partial t} = D_1 \frac{\partial^2 v_1}{\partial x^2} + D_1 \frac{\partial^2 v_1}{\partial y^2} + v_1 - v_1^3 - k - v_2,$$
$$\frac{\partial v_2}{\partial t} = D_2 \frac{\partial^2 v_2}{\partial x^2} + D_2 \frac{\partial^2 v_2}{\partial y^2} + v_1 - v_2, \tag{16}$$

where $x, y \in (-1,1)$, $t \in (0,5]$, $k = 0.005$, and the diffusion coefficients $D_1 = 0.001$ and $D_2 = 0.005$. The initial condition is generated from the standard Gaussian distribution. The numerical simulation process is detailed in (Takamoto et al., 2022). We are interested in four functions: $f_1 = v_1(2.5, x, y)$, $f_2 = v_2(2.5, x, y)$, $u_1 = v_1(5.0, x, y)$ and $u_2 = v_2(5.0, x, y)$. We extracted each function instance from a $64 \times 64$ sub-mesh from the numerical solutions.

### A.4    Torus Fluid

We considered a vicous, incompressive fluid on the unit torus. The governing equation in vorticity form is given by

$$\frac{\partial w(\mathbf{x},t)}{\partial t} + \mathbf{u} \cdot \nabla w(\mathbf{x},t) = \nu \nabla^2 w(\mathbf{x},t) + f(\mathbf{x}), \tag{17}$$
$$w(\mathbf{x},0) = w_0(\mathbf{x}), \tag{18}$$

---

[1] https://www.mathworks.com/help/matlab/math/partial-differential-equations.html

where $\mathbf{x} \in [0,1]^2$, $t \in [0,10]$, $\nu = 0.001$, $\mathbf{u}$ is the velocity field that $\nabla \cdot \mathbf{u} = 0$, $w(\mathbf{x}, t)$ is the vorticity function, and $f(\mathbf{x})$ is the source function that represents the external forces. We are interested in seven spatial functions: the initial condition $w_0$, the source function $f$, and the vortocity fields at time steps $t = 2, 4, 6, 8, 10$, denoted as $w_1$, $w_2$, $w_3$, $w_4$, and $w_5$. We employed a parametric form for $w_0$ and $f$: $w_0(\mathbf{x}) = [\sin(\alpha_1 \pi(x_1 + \beta_1)), \sin(\alpha_2 \pi(x_1 + \beta_2))] \cdot \mathbf{\Lambda} \cdot [\cos(\alpha_1 \pi(x_2 + \beta_1)), \cos(\alpha_2 \pi(x_2 + \beta_2))]^T$, where $\alpha_1, \alpha_2 \sim \text{Uniform}(0.5, 1)$, $\beta_1, \beta_2 \sim \text{Uniform}(0, 1)$, and each $[\mathbf{\Lambda}]_{ij} \sim \text{Uniform}(-1, 1)$, and $f(x_1, x_2) = 0.1 [a \sin(2\pi(x_1 + x_2 + c)) + b \cos(2\pi(x_1 + x_2 + d))]$ where $a, b \sim \text{Uniform}(0, 2)$ and $c, d \sim \text{Uniform}(0, 0.5)$. We used the finite-difference solver as provided in (Li et al., 2020b) to obtain the solution of $\{w_1, w_2, w_3, w_4, w_5\}$. The spatial domain was discretized to a $128 \times 128$ mesh. Again, to prepare the training and test datasets, we extracted the function instances from a $64 \times 64$ sub-mesh.

# B  Hyperparameter Selection

In the experiment, we used the validation dataset to determine the optimal hyperparameters for each method. The set of the hyperparameters and their respective ranges are listed as follows.

- FNO: the hyperparameters include the number of modes, which varies from {12, 16, 18, 20, 24}, the number of channels for channel lifting, which varies from {64, 128, 256}, and the number of Fourier layers, which varies from {2, 3, 4, 5}. We used GELU activation, the default choice in the original FNO library[2].

- DON: the hyperparameters include the number of convolution layers in the branch net, which varies from {3, 5, 7}, the kernel size in the convolution layer, which varies from {3, 5, 7}, the number of MLP layers in the truck net, which varies from {3, 4, 5}, the output dimension of the branch net and trunk net, which varies from {64, 128, 256}, and the activation, which varies from {ReLU, Tanh}.

- PODDON: the hyperparameters include the number of bases, varying from {128, 256, 512}, the number of convolution layers in the branch net, varying from {3, 5, 7}, the kernel size, varying from {3, 5, 7}, the output dimension of the branch net and trunk net, varying from {64, 128, 256}, and the activation, varying from {ReLU, Tanh}.

- GNOT: the hyperparameters include the number of attention layers, varying from {3, 4, 5}, the dimensions of the embeddings, varying from {64, 128, 256}, and the inclusion of mixture-of-expert-based gating, specified as either {yes, no}. We used GeLU activation, the default choice of the original library[3].

- Simformer (Gloeckler et al., 2024): the original Simformer was developed and tested on a small number of tokens. To handle a large number of sampling locations, which in our experiments amounts to $64 \times 64 = 4,096$, we employed the linear attention mechanism as used in GNOT. The number of attention layers was varied from {3,4, 5}. The dimensions of the embeddings were selected from {128, 256, 512}. The activation was selected from {GELU, ReLU, Tanh}.

- ACM-FD: the hyperparameters include the number of modes, which varies from {12, 16, 18, 20, 24}, the number of channels for channel lifting, which varies from {64, 128, 256}, the number of Fourier layers from, which varies from {3, 4, 5}, the length-scale of the SE kernel, which varies from {1e-2, 5e-3, 1e-3, 5e-4, 1e-4}. We used GELU activation.

- $\beta$-VAE: the hyperparameters include $\beta$, varying from {1e1, 1e-1, 1e-3, 1e-4, 1e-5, 1e-6}, the rank, varying from {16, 32, 64, 128, 256}, and the number of convolution layers in the decoder and encoder networks, varying from from {1, 2, 3, 4}. We employed the GELU activation and fixed the kernel size to be 3.

For FNO, DON, PODDON and GNOT, each task underwent an independent hyperparameter tuning process to identify the optimal hyperparameters specific to that task. In other words, each model was retrained from scratch for each individual task. For example, solving ten tasks would result in ten distinct FNO models. In contrast, for Simformer and ACM-FD, the model was trained only once, where the validation error is defined as the summation of the relative $L_2$ error across all the tasks. The same model is used for inference on every task. Note that, $\beta$-VAE is only trained for data generation (not for any prediction ask). The tuning of $\beta$-VAE is guided by the reconstruction error on the validation dataset.

---

[2]https://github.com/neuraloperator/neuraloperator
[3]https://github.com/HaoZhongkai/GNOT

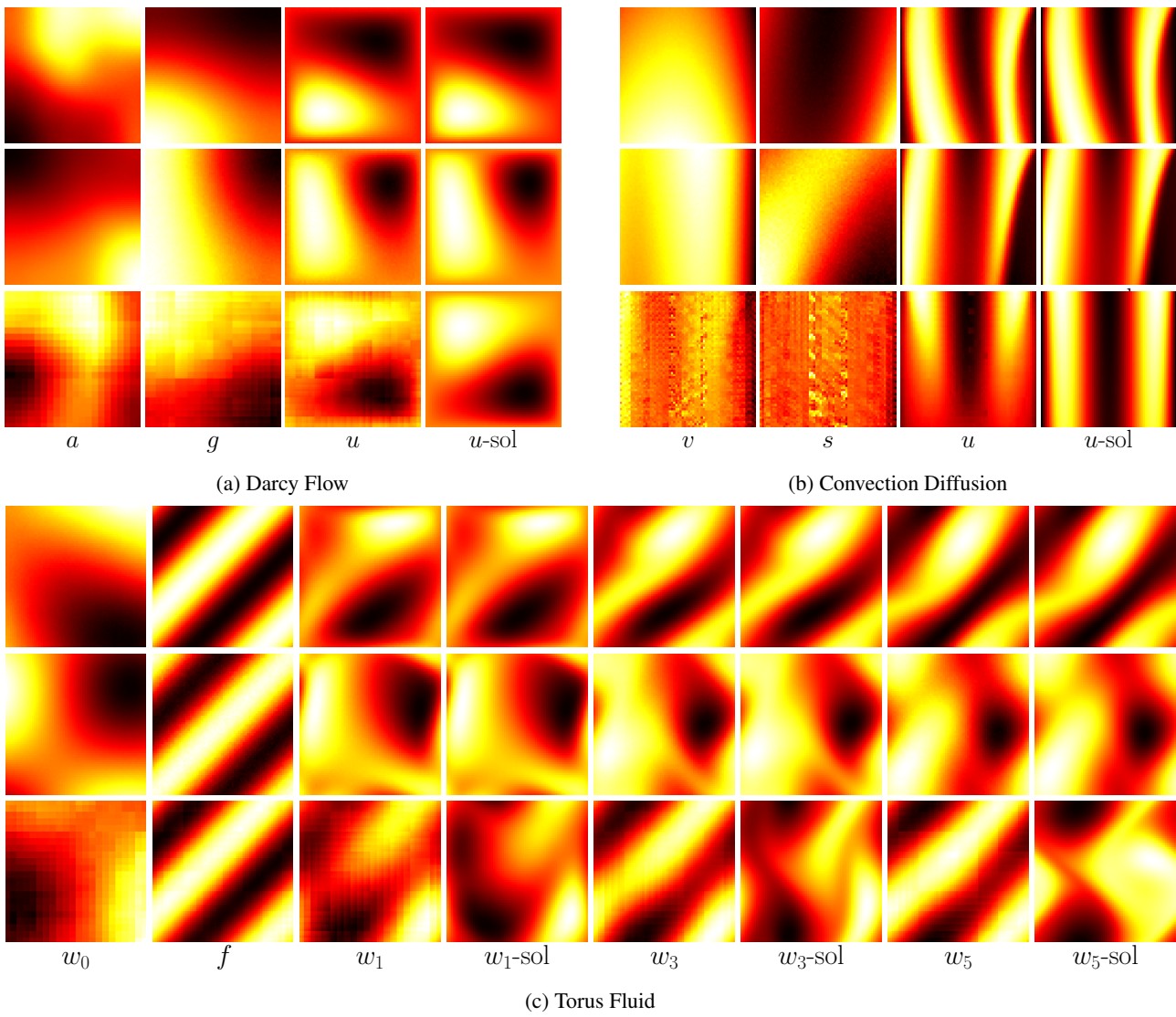

$$a \qquad g \qquad u \qquad u\text{-sol} \qquad\qquad v \qquad s \qquad u \qquad u\text{-sol}$$

(a) Darcy Flow                                    (b) Convection Diffusion

$$w_0 \qquad f \qquad w_1 \qquad w_1\text{-sol} \qquad w_3 \qquad w_3\text{-sol} \qquad w_5 \qquad w_5\text{-sol}$$

(c) Torus Fluid

*Figure 3.* Function instances generated by ACM-FD (top row), by MFD (middle row), and by $\beta$-VAE (third row). "-sol" means the numerical solution provided by the numerical solvers given the other functions.

## C   Multi-physics System Generation

We investigated the functions generated by each method. Figure 3 illustrates a set of functions randomly generated by ACM-FD, MFD, and $\beta$-VAE for Darcy Flow, Convection Diffusion and Torus Fluid systems. More examples are provided in Figure 4 and 5. As we can see, the generated solutions from both ACM-FD and MFD are highly consistent with the numerical solutions, accurately capturing both the global structures and finer details. In contrast, the solutions produced by $\beta$-VAE often fail to capture local details (see Fig. 3a for $u$ and Fig. 3c for $w_1$) or significantly deviate from the numerical solution in terms of overall shape (see Fig. 3b for $u$ and Fig. 3c for $w_3$). In addition, other functions generated by $\beta$-VAE, such as $(a, g)$ for Darcy Flow and $(v, s)$ for Convection Diffusion, appear quite rough and do not align well with the corresponding function families (see Section A). This might be due to that $\beta$-VAE is incapable of capturing the correlations between function values across different input locations. In contrast, ACM-FD introduces GP noise functions and performs diffusion and denoising within the functional space. As a result, ACM-FD is flexible enough to effectively capture the diverse correlations between function values. Overall, these results confirm the advantages of our method in multi-physics

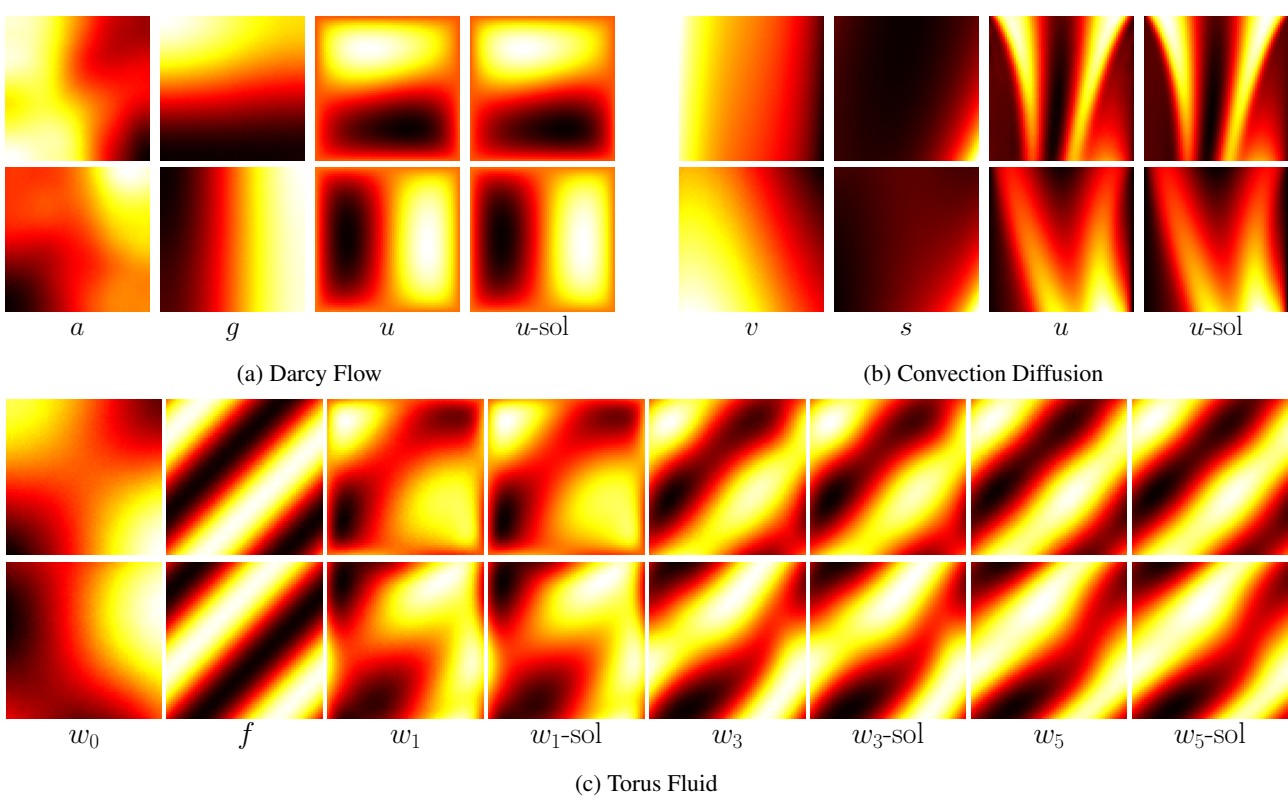

(a) Darcy Flow

(b) Convection Diffusion

(c) Torus Fluid

*Figure 4.* More generated function instances by ACM-FD. "-sol" means the numerical solution provided by the numerical solvers given the other functions.

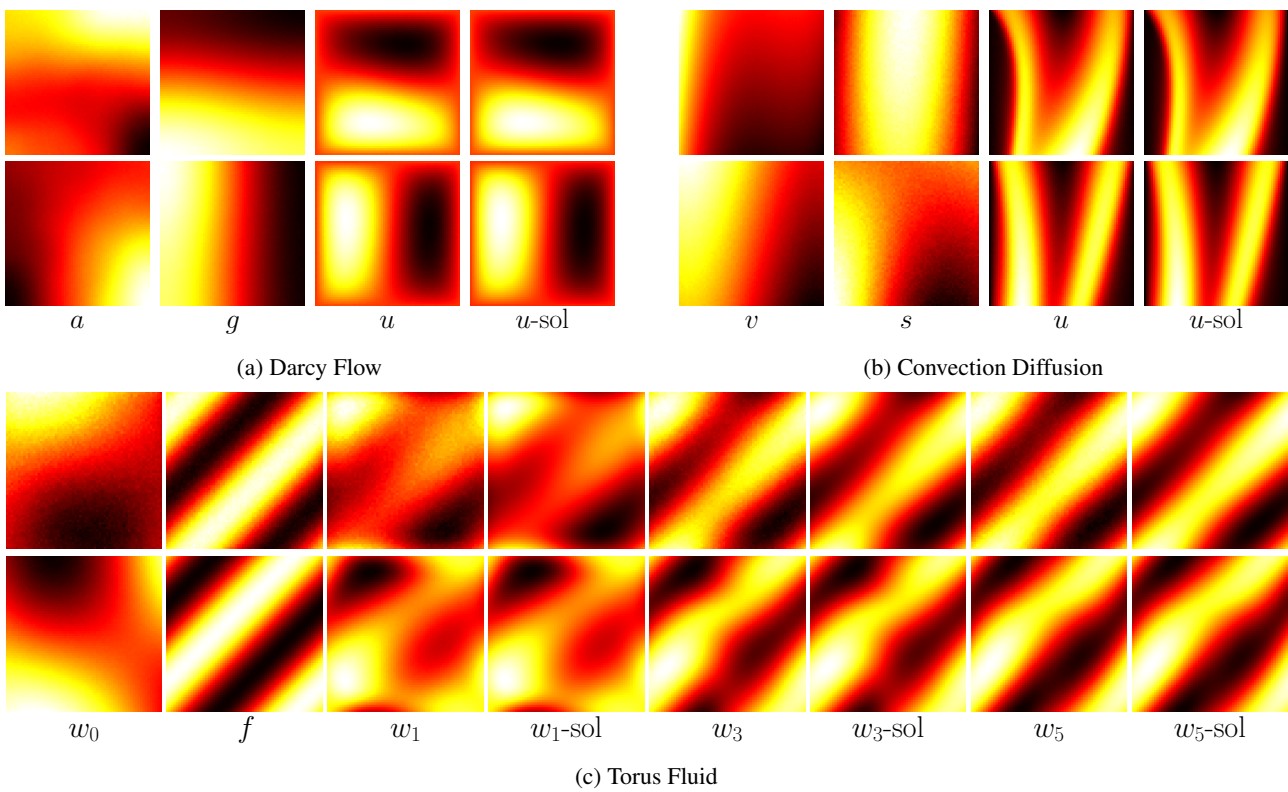

(a) Darcy Flow

(b) Convection Diffusion

(c) Torus Fluid

*Figure 5.* More generated function instances by MFD. "-sol" means the numerical solution provided by the numerical solvers given the other generated functions.

data generation.

*Table 5.* Relative $L_2$ error of ACM-FD *vs.* PODDON for various prediction tasks. The results were averaged over five runs. D-F, C-D and T-F are short for Darcy Flow, Convection Diffusion, and Torus Fluid Systems.

| Dataset | Task(s) | ACM-FD | PODDON |
|---|---|---|---|
| D-F | $f, u$ to $a$ | **1.32e-02 (2.18e-04)** | 1.56e-02 (1.44e-04) |
| | $a, u$ to $f$ | **1.59e-02 (1.59e-04)** | 4.63e-02 (1.21e-03) |
| | $a, f$ to $u$ | **1.75e-02 (4.16e-04)** | 6.80e-02 (2.16e-04) |
| | $u$ to $a$ | **3.91e-02 (7.08e-04)** | 4.09e-02 (4.18e-04) |
| | $u$ to $f$ | **3.98e-02 (6.45e-04)** | 6.37e-02 (1.03e-03) |
| C-D | $s, u$ to $v$ | **2.17e-02 (4.53e-04)** | 4.62e-02 (2.00e-04) |
| | $v, u$ to $s$ | **5.45e-02 (1.40e-03)** | 7.57e-02 (4.09e-04) |
| | $v, s$ to $u$ | **1.60e-02 (2.15e-04)** | 1.72e-01 (1.17e-03) |
| | $u$ to $v$ | **2.66e-02 (3.08e-04)** | 5.38e-02 (5.29e-04) |
| | $u$ to $s$ | **6.06e-02 (2.54e-04)** | 1.03e-01 (1.29e-03) |
| D-R | $f_1, u_1$ to $f_2$ | **1.44e-02 (8.96e-04)** | 3.85e-01 (2.00e-04) |
| | $f_1, u_1$ to $u_2$ | **1.59e-02 (3.68e-04)** | 2.67e-01 (4.09e-04) |
| | $f_2, u_2$ to $f_1$ | **4.10e-02 (8.93e-04)** | 5.09e-01 (1.17e-03) |
| | $f_2, u_2$ to $u_1$ | **5.86e-02 (3.43e-04)** | 3.70e-01 (5.29e-04) |
| T-F | $w_0, w_5$ to $w_1$ | **2.73e-02 (4.78e-03)** | 6.06e-02 (2.91e-04) |
| | $w_0, w_5$ to $w_2$ | **2.43e-02 (1.60e-03)** | 7.71e-02 (1.63e-04) |
| | $w_0, w_5$ to $w_3$ | **2.43e-02 (3.17e-03)** | 7.38e-02 (2.92e-04) |
| | $w_0, w_5$ to $w_4$ | **1.68e-02 (1.81e-03)** | 5.38e-02 (2.18e-04) |
| | $w_0, w_5$ to $f$ | **1.63e-02 (1.49e-03)** | 4.94e-02 (9.02e-04) |
| | $w_0, f$ to $w_1$ | **3.10e-02 (4.08e-03)** | 5.54e-02 (6.45e-04) |
| | $w_0, f$ to $w_2$ | **3.28e-02 (4.79e-03)** | 7.40e-02 (2.01e-04) |
| | $w_0, f$ to $w_3$ | **3.49e-02 (2.38e-03)** | 8.60e-02 (5.16e-04) |
| | $w_0, f$ to $w_4$ | **3.34e-02 (3.87e-03)** | 9.74e-02 (4.21e-04) |
| | $w_0, f$ to $w_5$ | **3.26e-02 (2.13e-03)** | 1.16e-01 (8.07e-04) |

## D   Ablation Studies

In this section, we conducted ablations studies to assess the effect of critical components of our method.

- **Random Masking.** First, we evaluated the importance of our random masking strategy in model training. To this end, we compared ACM-FD with MFD — the variant that was trained without random masks. That is, the training is fully unconditional. Table 6 shows the relative $L_2$ error in D-F and C-D systems. As we can see, across various prediction tasks, our model trained with random masks consistently outperforms the variant without them, achieving substantial error reductions ranging from 40.9% to 96.2%.

- **Choice of Masking Probability.** Next, we investigated the impact of the masking probability $p$. In the main paper, all results were obtained using a neutral setting of $p = 0.5$, which yields the highest masking variance. Here, we varied $p$ across $\{0.2, 0.4, 0.6, 0.8\}$ to evaluate the prediction accuracy of our method on the D-F and C-D systems. As shown in Table 7, the performance at $p = 0.4$ and $p = 0.6$ remains comparable to that at $p = 0.5$, indicating a degree of robustness to the choice of $p$. However, more extreme values, such as $p = 0.2$ or $p = 0.8$, lead to a noticeable drop in performance.

- **Kronecker Product based Computation.** Third, we evaluated the impact of incorporating the Kronecker product in our method and compared both training and inference time with other approaches. All runtime experiments were conducted on a Linux cluster node equipped with an NVIDIA A100 GPU (40GB memory).

  To ensure a fair and comprehensive comparison, we first measured the per-epoch training time. For neural operator baselines, we recorded the total per-epoch time across all the tasks. The results, as summarized in Table 8, show that leveraging Kronecker product properties greatly improves the training efficiency of our method. Our per-epoch training time is substantially lower than that of all competing methods except DON.

  However, diffusion-based models typically require far more training epochs than deterministic neural operators. For example, FNO and GNOT converge within 1K epochs in all the settings, while our method generally requires around

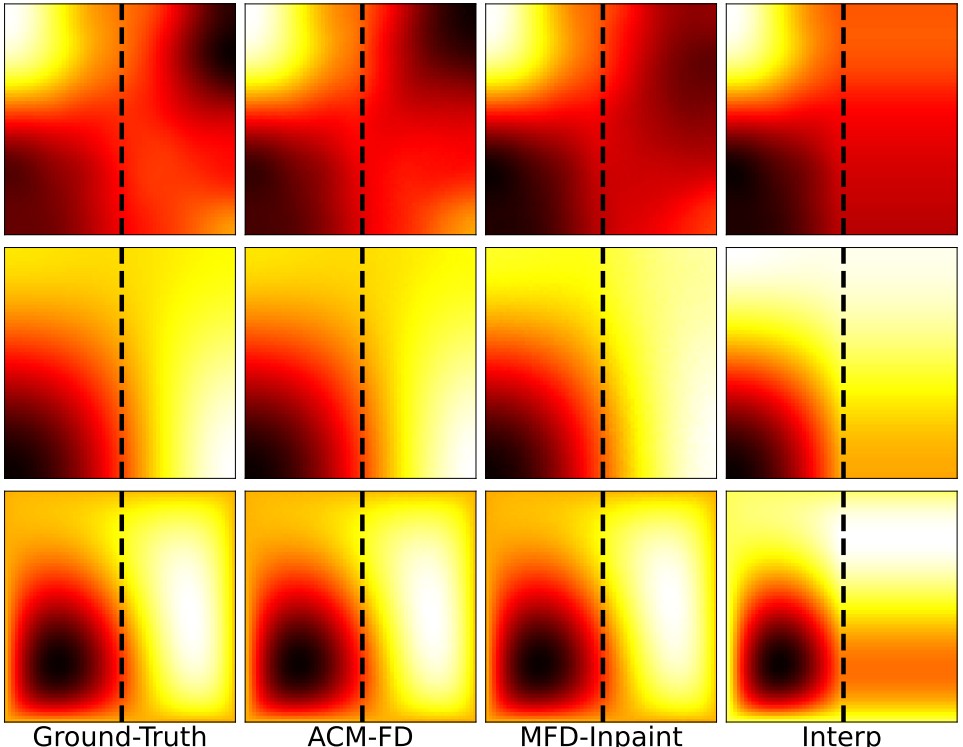

(a) Darcy Flow: examples of completing functions $a$ (the first row), $f$ (the second row) and $u$ (the third row).

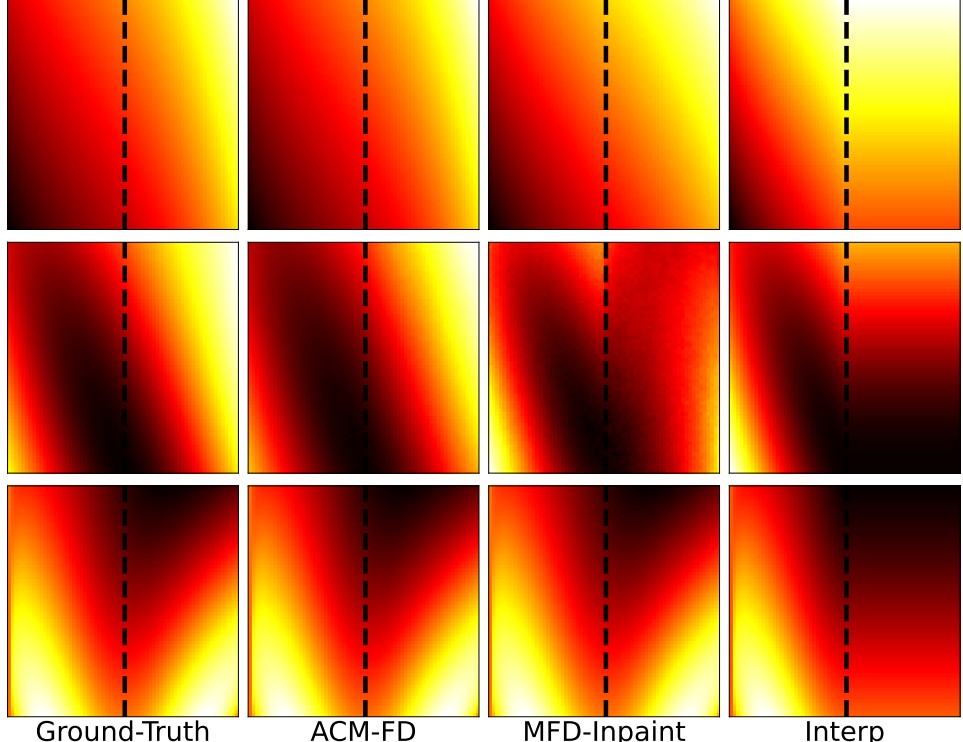

(b) Convection Diffusion: examples of completing functions $v$ (the first row), $s$ (the second row), and $u$ (the third row).

*Figure 6.* Examples of function completion. The functions are sampled from the left half of the domain, and the right half are completed by each method.

20K epochs. As a result, despite the per-epoch efficiency, the total training time for our method — as well as for Simfomer, another diffusion-based model — remains higher overall.

Lastly, by using Kronecker product, our method achieves substantial acceleration in generation and/or prediction, with a 7.4x speed-up on C-D and a 6.9x speed-up on D-F. During inference, the computational cost is dominated by sampling noise functions, whereas during training, a substantial portion of cost arises from gradient computation. Consequently, the runtime advantage of using the Kronecker product is even more pronounced during inference.

*Table 6.* Comparison of MFD and ACM-FD in relative $L_2$ errors.

| Dataset | Task | MFD | ACM-FD |
|---|---|---|---|
| D-F | $f, u$ to $a$ | 1.70e-1 (3.45e-3) | **1.32e-2 (2.18e-4)** |
| | $a, u$ to $f$ | 6.98e-2 (3.09e-3) | **1.59e-2 (1.59e-4)** |
| | $a, f$ to $u$ | 2.96e-2 (1.16e-3) | **1.75e-2 (4.16e-4)** |
| | $u$ to $a$ | 1.70e-1 (3.56e-3) | **3.91e-2 (7.08e-4)** |
| | $u$ to $f$ | 1.05e-1 (4.30e-3) | **3.98e-2 (6.45e-4)** |
| C-D | $s, u$ to $v$ | 5.47e-1 (3.56e-2) | **2.17e-2 (4.53e-4)** |
| | $v, u$ to $s$ | 3.95e-1 (4.01e-2) | **5.45e-2 (1.40e-3)** |
| | $v, s$ to $u$ | 3.68e-2 (1.65e-3) | **1.60e-2 (2.15e-4)** |
| | $u$ to $v$ | 6.94e-1 (3.64e-2) | **2.66e-2 (3.08e-4)** |
| | $u$ to $s$ | 9.23e-1 (3.64e-2) | **6.06e-2 (2.54e-4)** |

*Table 7.* Relative $L_2$ error across different masking probabilities $p$.

| Dataset | Task | $p$=0.2 | 0.4 | 0.5 | 0.6 | 0.8 |
|---|---|---|---|---|---|---|
| D-F | $f, u$ to $a$ | 2.16e-2 | 1.60e-2 | **1.32e-2** | 1.34e-2 | 1.26e-2 |
| | $a, u$ to $f$ | 1.85e-2 | 1.61e-2 | **1.59e-2** | 1.67e-2 | 1.70e-2 |
| | $a, f$ to $u$ | 2.50e-2 | 1.95e-2 | **1.75e-2** | 2.05e-2 | 2.00e-2 |
| | $u$ to $a$ | 4.48e-2 | 4.14e-2 | **3.91e-2** | 3.93e-2 | 4.96e-2 |
| | $u$ to $f$ | 4.26e-2 | 4.07e-2 | **3.98e-2** | 4.32e-2 | 5.02e-2 |
| C-D | $s, u$ to $v$ | 3.24e-2 | 2.72e-2 | **2.17e-2** | 2.33e-2 | 2.91e-2 |
| | $v, u$ to $s$ | 7.11e-2 | 6.85e-2 | **5.45e-2** | 5.86e-2 | 8.35e-2 |
| | $v, s$ to $u$ | 1.81e-2 | 1.76e-2 | **1.60e-2** | 1.77e-2 | 3.56e-2 |
| | $u$ to $v$ | 2.91e-2 | **2.41e-2** | 2.66e-2 | 2.65e-2 | 5.15e-2 |
| | $u$ to $s$ | 7.69e-2 | **5.62e-2** | 6.06e-2 | 6.91e-2 | 9.15e-2 |

*Table 8.* Comparison of training and inference time on C-D and D-F systems. ACM-FD (w/o K) means running our method without using Kronecker product properties for computation.

(a) Training time per epoch (in seconds)

| Dataset | ACM-FD | ACM-FD (w/o K) | Reduction | Simformer | FNO | GNOT | DON |
|---------|--------|----------------|-----------|-----------|-----|------|-----|
| C-D | 3.09 | 5.5 | 43.8% | 23.6 | 16.45 | 144 | 1.53 |
| D-F | 3.27 | 5.55 | 41.4% | 23.5 | 15.6 | 148 | 1.48 |

(b) Total training time (in hours)

| Dataset | ACM-FD | ACM-FD (w/o K) | Simformer | FNO | GNOT | DON |
|---------|--------|----------------|-----------|-----|------|-----|
| C-D | 17.2 | >48 | 45.9 | 4.57 | 40 | 4.25 |
| D-F | 18.2 | >48 | 45.7 | 4.33 | 41.1 | 4.11 |

(c) Inference time per sample (in seconds)

| Dataset | ACM-FD | ACM-FD (w/o K) | Reduction | Simformer |
|---------|--------|----------------|-----------|-----------|
| C-D | 0.899 | 6.66 | 86.5% | 7.39 |
| D-F | 0.975 | 6.7 | 85.4% | 7.34 |

