# OpenReview forum: "Arbitrarily-Conditioned Multi-Functional Diffusion for Multi-Physics Emulation"
_ICML.cc/2025/Conference — ICML 2025 poster_

### Official Review · Reviewer_hz2Z · 2025-02-14

**Overall Recommendation:** 4

**Summary:**

The authors outline a generative modeling technique to produce functional solutions to multi-function problems, with an emphasis on emulating multi-function physical systems. This is accomplished by casting the typical DDPM framework into the functional space, where the noise is framed as a Gaussian process. Then, DDPM effectively samples at sampling points along the functions domain, and this is extended to multiple functions simultaneously. Key contributions also include a tailored loss function for arbitrary conditioning, including conditioning on entire functions or on subsets of particular functional domains. They also provide an efficient scheme for sampling from the GP. They compare to extensive state-of-the art baselines, including neural operators and a diffusion-based method, exhibiting superior performance. In an ablation-like study, they verify that the explicit training using the masked loss and 0-filling is beneficial.

## update after rebuttal: I believe my original score holds, and that this work is a good contribution.

**Claims And Evidence:**

Their main claim is the advantage and flexibility of the arbitrary conditioning in generative models when solving diverse inverse problems. Training to solve all functions simultaneously, the authors demonstrate that their method outperforms methods trained specifically for each task individually, validating that the multi-function estimation approach is advantageous.

**Essential References Not Discussed:**

While not necessary for comparison in experiments, it might be good to acknowledge other diffusion-based methods for solving inverse problems, as that is a major focus of this work. The most important work in this domain is perhaps [1], with others including [2,3]. These methods need not be compared with experimentally, but it would be interesting to discuss the advantages and differences between these approaches and the outlined approach in the context of multi-function generation and inverse problems.

[1] Chung et al. "Diffusion Posterior Sampling for General Noisy Inverse Problems." ICLR 2023.
[2] Wang et al. "Zero-Shot Image Restoration Using Denoising Diffusion Null-Space Model." ICLR 2023.
[3] Song et al. "Pseudoinverse-guided diffusion models for inverse problems." ICLR, 2023.

**Experimental Designs Or Analyses:**

The experiments sufficiently assess the validity of the approach. Natural questions might include (1) are neural operators trained for specific tasks more effective, (2) are other diffusion-like approaches more effective, particularly those with different conditioning strategies, and (3) how effective is the uncertainty quantification enabled by the method? The authors clearly and effectively answer questions (1) and (2) in the experiments, evaluating on a wide array of settings with competitive baselines. However, despite mentioning uncertainty quantification earlier in the paper, it is not addressed in the experiments. It would be interesting to see how uncertainty intervals or estimates obtained from this method compare to those obtained from other capable methods, such as simformer.

**Methods And Evaluation Criteria:**

The authors rigorously evaluate their methods using L2 error on the reconstruction of functional outputs. Additionally, they assess the ability of their method to align with true physical equations, which is a particularly important consideration and often overlooked. Similarly, they assess the diversity in their outputs (exhibiting competitive performance). Finally, they fully examine the conditional capability of their approach for full-function prediction (forward models and inverse problems) and interpolation within a given function’s domain.

**Other Comments Or Suggestions:**

The position of algorithm 1 in the middle of the text is a bit strange, maybe it would be better to move it to the top of the page.

**Other Strengths And Weaknesses:**

I think this a strong work with rigorous experimental comparisons. While I was not initially convinced in the main text by the utility of the masked loss function for training a diffusion model capable of flexible conditional generation, it is made evident in the experiments that this is indeed necessary, as it is demonstrated to be more effective than prior methods with trained conditioning and an “ablated” version of the proposed approach. Moreover, the authors focus on key important evaluation metrics, such a physical consistency and diversity, which is an often overlooked and important component in generative modeling for surrogate modeling, solving inverse problems, and unconditional generation. The main downside, however, is that the authors did not further investigate the ability of their approach for uncertainty quantification, which, as they note, is another important consideration to be made for modeling physical systems. With that said, I believe they have outlined an approach that will be valuable to share with the community.

**Questions For Authors:**

1.The arbitrary choice of h being Bernoulli-distributed with p=0.5 is not justified. Wouldn’t a network trained with such an assumption always “assume” the input to have roughly half of the components masked? Have you experimented with varying p through training?

2. “We used FNO to construct our denoising network” What does this mean? You share the same architecture as FNO?

**Relation To Broader Scientific Literature:**

The authors position their work within the context of generative modeling via diffusion. Specifically, they connect to other diffusion-based methods for functional generation/generating the outputs to a function, making the distinction that their work focuses on the generation of multiple related functions simultaneously. They also connect to the neural operator literature, which aims to learn function-to-function mappings, e.g., corresponding to solutions of PDEs. However, by leveraging diffusion/generative modeling, their framework naturally incorporates uncertainty quantification in addition to prediction.

**Theoretical Claims:**

The authors do not make theoretical claims.

---

> ### Author Rebuttal · Authors · 2025-04-01
>
> We thank the reviewer for their insightful comments and valuable suggestions.
>
> > It would be interesting to see how uncertainty intervals or estimates obtained from this method compare to those obtained from other capable methods, such as simformer.
>
> R1: Thank you for the great suggestion — we completely agree. We have added the evaluation and analysis of uncertainty quantification. Please see our **response R1 to Reviewer bmeB** for details.
>
> >The most important work in this domain is perhaps [1], with others including [2,3]. These methods need not be compared with experimentally, but it would be interesting to discuss the advantages and differences between these approaches
>
> R2: Thank you very much for providing these excellent references! We will certainly cite and discuss them.
>
> The key difference lies in the training and inference paradigms. The referenced works [1–3] focus on training an **unconditional diffusion model**, and only at inference time do they incorporate the likelihood of observations or measurements to guide sampling from the conditional distribution. The advantage of this approach is that it only requires a standard diffusion model --- potentially even a pre-trained one --- without the need to address conditional sampling tasks during training. Indeed, our baseline **MFD-Inpaint** (see Section 5.2) follows this paradigm.
>
> However, the referenced methods may have limitations when applied to our scenarios:[1] requires a closed-form expression of the likelihood or forward model $p(y|x_0)$, or at least an efficient way to compute its gradient (or sensitivity) --- which is often infeasible in multi-physics systems governed by complex differential equations. The work [2] further restricts the applicability by assuming that the forward model is linear. [3] is more flexible and can handle nonlinear forward models, but it requires a "pseudo-inverse" operator for the forward process, which might not be easy to obtain.
>
>  Our approach takes a different route: during training, we **explicitly incorporate conditionality** by randomly sampling masks and training our diffusion model to handle a wide range of conditional sampling tasks (including the unconditional case). At inference time, we **do not incorporate any likelihood scores** --- instead, we directly sample from our learned denoising model. Empirically, we found that this conditional training strategy yields **large improvements over unconditional training approaches** like MFD-Inpaint across a variety of tasks (see Table 3 and our response R1 to Reviewer BtjJ).
>
> > The arbitrary choice of h being Bernoulli-distributed with p=0.5 is not justified... Have you experimented with varying p through training?
>
> R3: Thank you for the insightful question. When there is no prior knowledge about whether a function value should be conditioned on or be generated, we suggested using $p = 0.5$ to avoid bias --- giving each component an equal chance of being conditioned or sampled. This is actually a uniform prior. We do agree, the mean of the proportion of the masked components is 0.5; but the standard deviation is also 0.5 ($\sqrt{p\times(1-p)}=0.5)$, which is the maximum possible, indicating that **the actual masked proportion can vary widely**. In our experiments, we used $p = 0.5$ to maintain a neutral setting. However, our method places no restrictions on the choice of $p$; it can be adjusted based on domain knowledge or specific application needs.
>
> We have now included results for alternative values of $p$: 0.2, 0.4, 0.6, and 0.8. As shown below, the performance for $p=0.4$ and $p=0.6$ is comparable to that of $p=0.5$, demonstrating a certain degree of robustness to the choice of $p$. However, when $p$ is set to more extreme values, such as 0.2 or 0.8, the performance degrades markedly.
>
> We will add these results and discussions into our paper.
>
> **Relative $L_2$ error**
> |System|Task|$p=0.2$|0.4|0.5|0.6|0.8|
> |-|-|-|-|-|-|-|
> DF|$f,u$ to $a$|2.16e-2|1.6e-2|**1.32e-2**|1.34e-2|1.26e-2|
> ||$a,u$ to $f$|1.85e-2|1.61e-2|**1.59e-2**|1.67e-2|1.7e-2|
> ||$a, f$ to $u$|2.5e-2|1.95e-2|**1.75e-2**|2.05e-2|2e-2|
> ||$u$ to $a$|4.48e-2|4.14e-2|**3.91e-2**|3.93e-2|4.96e-2|
> ||$u$ to $f$|4.26e-2|4.07e-2|**3.98e-2**|4.32e-2|5.02e-2|
> CD|$s,u$ to $v$|3.24e-2|2.72e-2|**2.17e-2**|2.33e-2|2.91e-2|
> ||$v,u$ to $s$|7.11e-2|6.85e-2|**5.45e-2**|5.86e-2|8.35e-2|
> ||$v,s$ to $u$|1.81e-2|1.76e-2| **1.60e-2**|1.77e-2|3.56e-2|
> ||$u$ to $v$|2.91e-2|**2.41e-2**| 2.66e-2|2.65e-2|5.15e-2|
> ||$u$ to $s$|7.69e-2|**5.62e-2**| 6.06e-2|6.91e-2|9.15e-2|
>
> > “We used FNO to construct our denoising network” What does this mean? You share the same architecture as FNO?
>
> R4: Great question. Yes, we use the same architecture as FNO - specifically, a sequence of Fourier layers — to construct our denoising network. We will make this point clearer in the paper.

---

> > ### Comment · Reviewer_hz2Z · 2025-04-05
> >
> > Thank you for addressing my concerns. I maintain that this paper represents a good contribution to the conference, and believe my original score holds.

---

> > > ### Author Response · Authors · 2025-04-06
> > >
> > > Thank you for your support and feedback to our response!

---

### Official Review · Reviewer_BtjJ · 2025-03-14

**Overall Recommendation:** 3

**Summary:**

The paper presents Arbitrarily-Conditioned Multi-Functional Diffusion (ACM-FD), a novel probabilistic surrogate model for multi-physics emulation. The key contributions include:

- A multi-functional diffusion framework based on DDPM that models noise as multiple Gaussian processes, enabling generation of multiple functions in multi-physics systems
- An innovative denoising loss using random masks that handles all possible conditional parts within the system
- An efficient training and sampling approach using Kronecker product structure in the GP covariance matrix
- Comprehensive experiments across four multi-physics systems showing superior performance compared to state-of-the-art neural operators

The paper demonstrates that ACM-FD can handle various tasks including forward prediction, inverse problems, completion, and data simulation within a single framework, while providing uncertainty quantification.

**Claims And Evidence:**

The paper's claims are generally well-supported by evidence, with a few areas that could benefit from additional clarification:

Strong Claims with Clear Evidence:
- The ability to handle multiple functions of interest
- The computational efficiency gains through Kronecker product structure
- The superior performance across 24 prediction tasks

Areas Needing Additional Support:
- The claim about "substantially reducing the training and sampling costs" would benefit from quantitative comparisons with baseline methods

**Essential References Not Discussed:**

N/A

**Experimental Designs Or Analyses:**

The experimental design is comprehensive but could be enhanced:

Suggestions for Improvement:
- Add statistical significance tests for the performance comparisons
- Include more details about the training data generation process
- Provide visualization of the generated functions and their uncertainty estimates

**Methods And Evaluation Criteria:**

Suggestions for Improvement:
- Consider adding ablation studies to demonstrate the importance of each component (random masks, Kronecker structure)
- There are also some diffusion model works [1,2] that add masks during the diffusion process, and the reviewers believe that this needs to be discussed. NOTE: these two works are also using diffusion models to inverse problem.
- Add more details about the training process and hyperparameter selection

[1] High-Frequency Space Diffusion Model for Accelerated MRI

[2] Measurement-conditioned denoising diffusion probabilistic model for under-sampled medical image reconstruction

**Other Comments Or Suggestions:**

N/A

**Other Strengths And Weaknesses:**

Weaknesses:
- Limited theoretical analysis of the proposed method
- Lack of comparison with traditional numerical methods
- Need for more detailed ablation studies
- Could benefit from more extensive visualization of results

**Questions For Authors:**

N/A

**Relation To Broader Scientific Literature:**

N/A

**Theoretical Claims:**

N/A

---

> ### Author Rebuttal · Authors · 2025-04-01
>
> We thank the reviewer for the many constructive comments.
>
> > Consider adding ablation studies to demonstrate importance of each component (random masks, Kronecker structure)
>
> R1: Great suggestion. Actually, we have already conducted studies on **the effect of random masks**. In Section 5.3, we compared our method (ACM-FD) with a variant trained **without** random masks (denoted as MFD-Inpaint) on the function completion task. As shown in Table 3, incorporating random masks during training significantly reduces the relative $L_2$ error --- by as much as 58\% to 96\%.
>
> Here, we additionally include results for prediction tasks, presented below. Across various prediction settings, our model trained with random masks consistently outperforms the variant without them (denoted as MFD), achieving substantial error reductions ranging from 40.9% to 96.2%.
>
> **Relative $L_2$ error**
> |System | Task  | MFD | ACM-FD  |
> |-|-|-|-|
> DF|$f,u$ to $a$|1.70e-1 (3.45e-3)|**1.32e-2 (2.18e-4)**|
> ||$a,u$ to $f$|6.98e-2 (3.09e-3)|**1.59e-2 (1.59e-4)**|
> ||$a, f$ to $u$|2.96e-2 (1.16e-3)|**1.75e-2 (4.16e-04)**|
> ||$u$ to $a$|1.70e-1 (3.56e-3)|**3.91e-2 (7.08e-04)**|
> ||$u$ to $f$ |1.05e-1 (4.3e-3)|**3.98e-2 (6.45e-04)**|
> CD|$s,u$ to $v$|5.47e-1 (3.56e-2)|**2.17e-2 (4.53e-04)**|
> ||$v,u$ to $s$|3.95e-1 (4.01e-2)|**5.45e-2 (1.40e-03)**|
> ||$v,s$ to $u$|3.68e-2 (1.65e-3)|**1.60e-2 (2.15e-04)**|
> ||$u$ to $v$|6.94e-1 (3.64e-2)|**2.66e-2 (3.08e-04)**|
> ||$u$ to $s$|9.23e-1 (3.64e-2)|**6.06e-2 (2.54e-04)**|
>
> We have also supplemented the ablation study on the use of the Kronecker product structure. Please **see R2 to Reviewer bme8** for details.
>
> > discuss [1,2]
>
> R2: Thank you for the excellent references. We will definitely cite and discuss them. A key distinction is that the masks used in [1,2] are **fixed** and derived from the sampling patterns inherent to the medical imaging data. As a result, these works focus on a **single** inverse task --- either recovering high-frequency regions [1] or unsampled measurements [2]. In contrast, our method **repeatedly** samples **random masks** during training, enabling it to jointly handle a wide variety of conditional sampling tasks, encompassing all kinds of forward prediction, inverse prediction, and completion tasks.
>
> Another critical difference lies in the **functional space formulation** of our model: the noises used during both training and inference are stochastic functions sampled from Gaussian processes. Our method is designed to generate functions either unconditionally or conditioned on other function samples. This motivation leads to fundamentally different designs in model architecture, training loss, and computational techniques (e.g., the use of Kronecker product).
>
> > Add more details about the training process and hyperparameter selection
>
> R3: Thank you for the great suggestion. We actually have provided detailed information in **Appendix Section B**, including the set of hyperparameters, their ranges, implementation details and libraries used for each method, as well as the validation settings. Due to space constraints, only a subset of this information is included in the main paper, with a reference to the appendix (see Line 366). In response to your feedback, we will further expand and enrich Appendix Section B to improve clarity and completeness.
>
> > Add statistical significance tests for the performance comparisons
>
> R4: Thank you for the excellent suggestion. Based on a $z$-test conducted on the prediction errors with associated error bars, our approach outperforms the baseline methods in the vast majority of cases at the 95% significance level. We will include the results in our paper.
>
> >Include more details about the training data generation process
>
> R5: We actually have included these details in **Appendix Section A**, which provides the governing equations, numerical simulation library used, simulation procedures, and data collection process. We believe this information is sufficient to generate all the data for our experimental settings. Additionally, we will publicly release our experimental data.
>
> > Provide visualization of the generated functions and their uncertainty estimates
>
> R6: Thank you for the great suggestion. We have added the suggested visualizations and analysis --- **please see our response R1 to Reviewer bmeB for details**.
>
> > Lack of comparison with traditional numerical methods
>
> R7: In fact, we did compare with traditional numerical methods, as all our **test data** (as well as the training data) are generated using them. We consider the outputs of the traditional methods as the **gold standard**, and our goal is to train a surrogate model that closely approximates their results. The relative $L_2$ error reported in Table 1-3 quantifies the difference of our method's prediction **as compared to the traditional methods**.

---

> > ### Comment · Reviewer_BtjJ · 2025-04-07
> >
> > Thanks to the authors for their responses during the rebuttal period. I now have a deeper understanding of the details in the paper, and I will accordingly raise my score. However, I still believe that the claim about 'substantially reducing the training and sampling costs' would benefit from quantitative comparisons with baseline methods, which do not appear to have been addressed in the rebuttal.

---

> > > ### Author Response · Authors · 2025-04-08
> > >
> > > Thank you for your response. We truly appreciate your positive feedback and additional suggestion!
> > >
> > > To clarify regarding the training and sampling costs: we have indeed provided **quantitative comparisons** with the baselines in **our response R2 to Reviewer bmeB** (they were not included again in this rebuttal thread due to space limit). By leveraging the Kronecker product structure, our method achieves substantial reductions in both training and inference time—over 40% in training and 85% in inference—as shown in the "Runtime reduction" column of the corresponding tables in R2 to Reviewer bmeB. Please see more details, discussion, and analysis in R2 to Reviewer bmeB.
> > >
> > > We will make sure to incorporate these results and the associated discussion in our paper.

---

### Official Review · Reviewer_bmeB · 2025-03-24

**Overall Recommendation:** 4

**Summary:**

The paper introduces Arbitrarily-Conditioned Multi-Functional Diffusion (ACM-FD), a novel probabilistic surrogate model designed for multi-physics emulation. It aims to address limitations of traditional machine learning-based surrogate models, which are typically task-specific and lack uncertainty quantification. ACM-FD is based on the Denoising Diffusion Probabilistic Model (DDPM) but extends it to handle multiple functions in multi-physics systems with a unified framework. The key contributions are:

1. Multi-Functional Diffusion Framework: The paper introduces a framework based on DDPM that uses Gaussian Processes (GPs) for noise modeling, enabling the generation of multiple functions in multi-physics systems.
2. Innovative Denoising Loss: A random-mask-based, zero-regularized denoising loss is proposed to handle conditional generation tasks, improving network stability and task flexibility.
3. Efficient Training and Sampling: The model reduces computational costs by leveraging a Kronecker product structure and tensor algebra to simplify GP covariance matrix operations.
4. Experiments: ACM-FD reaches top-tier performance in 24 prediction tasks across four multi-physics systems, achieving high accuracy, adherence to governing equations, and diversity in generated data. It also excels in function completion compared to inpainting and interpolation methods.

**Claims And Evidence:**

Most of the claims made in this paper are well supported by clear evidence.

However, the authors have highlighted the importance of uncertainty quantification (UQ) and the advantage of their proposed method over the baselines by diffusion model naturally supporting UQ; while there is no analysis presented for UQ capabilities (i.e. ECE scores / confidence intervals that match the stochasticity in the data if any).

The authors are also claiming that the proposed method is efficient to train with Kronecker product, it would be supportive to have a training time comparison table to demonstrate the advantage. Since ACM-FD only needs to be trained once for each dataset, it would be reasonable to compare the total time of training other baselines for all prediction tasks regarding a dataset with the time ACM-FD needs.

**Essential References Not Discussed:**

N/A

**Experimental Designs Or Analyses:**

The authors compare ACM-FD against various baseline models, such as Fourier Neural Operator (FNO), Transformer-based Neural Operator (GNOT), DeepONet (DON), and POD-based DON (PODDON). These baselines are compared thoroughly across 4 Physical systems: Darcy Flow, Convection-Diffusion, Diffusion-Reaction, and Torus Fluid Systems; together with 4 prediction tasks for each system: forward prediction, inverse inference, joint simulation, and function completion.

ACM-FD outperforms other baselines in most of the tasks. But FNO outperforms ACM-FD in 8 out of 10 Torus Fluid Systems tasks, resembling potential improvements for ACM-FD.

**Methods And Evaluation Criteria:**

The proposed methods and evaluation criteria are suitable for the problem presented. However, the uncertainty quantification performance is not evaluated although demonstrated as an advantage over other baseline models in the paper.

**Other Comments Or Suggestions:**

It would be clearer to have another figure to demonstrate how the inputs and outputs of the target function are discretized on the mesh.

**Other Strengths And Weaknesses:**

Strengths:

1. Paper is well written and easy to follow.
2. The random masking and function discretization together provides a simple but efficient way to learn the full functional space (from data) during training.
3. The proposed method is novel in Multi-Physics Emulation field and eliminates the need to retrain models for diverse prediction tasks.
4. Empirical results show strong performance improvements.

Weaknesses:

1. Although authors have demonstrated the natural uncertainty quantification advantage with DDPM compared to the baselines, there is no analysis of UQ performance.
2. Efficiency in training the DDPM model is also presented as a contribution, which lacks a training-time comparison against baseline models. For details of suggestion please refer to "Claims And Evidence" section in the review.

**Questions For Authors:**

Please refer to weaknesses.

**Relation To Broader Scientific Literature:**

ACM-FD advances beyond the baseline methods by enabling a single model to perform forward prediction, inverse inference, joint simulation, and function completion, with built-in uncertainty quantification. Works like PODDON have sought to improve efficiency in neural operators but still fall short in handling diverse prediction tasks. The generalization capability to different sub-tasks from one unified training stage can have great impact in reducing physics emulation costs.

**Theoretical Claims:**

In this work, the authors focus on the methodological framework with empirical validation.

---

> ### Author Rebuttal · Authors · 2025-04-01
>
> We sincerely thank the reviewer for the thoughtful and constructive comments.
>
> > no analysis of UQ performance.
>
> R1: Great suggestion. Here we add UQ performance evaluation & analysis results.
>
> First, to evaluate UQ quality, we computed the **emprical coverage probability** [1]: $CP = \frac{1}{N} \sum_{i=1}^N I(y_i \in C_\alpha)$ where $y_i$ is the ground-truth function value, and $C_\alpha$ is the $\alpha$ confidence interval derived from 100 predictive samples generated by our method. We varied $\alpha$ from {90%, 95%, 99%}, and examined our method in eight prediction tasks across Convection-Diffusion (C-D) and Darcy Flow (D-F) systems. We compared against Simformer. Note that the other baselines are deterministic and unable to perform UQ.
>
> **C-D**
> | Task  | Method | $\alpha$=0.9   | 0.95 |  0.99|
> |-|-|-|-|-|
> | $s,u$ to $v$ | Ours | **0.833**  | **0.88**  | **0.921**   |
> |  | Simformer | 0.736  | 0.814  | 0.871   |
> | $v,u$ to $s$ | Ours | **0.766**  | **0.842**  | **0.913**   |
> |  | Simformer | 0.683  | 0.767  | 0.879   |
> | $v,s$ to $u$ | Ours | **0.939**  | **0.968**  | **0.99**   |
> |  | Simformer | 0.695  | 0.771  | 0.858   |
> | $u$ to $v$ | Ours | **0.821**  | **0.87**  | **0.922**   |
> |  | Simformer | 0.775  | 0.85  | 0.912   |
> | $u$ to $s$ | Ours | **0.92**  | **0.949**  | **0.972**   |
> |  | Simformer | 0.716  | 0.773  | 0.823   |
>
> **D-F**
> | Task  | Method | $\alpha$=0.9   | 0.95 |  0.99|
> |----------|-----------|--------------------|----------------------------|-----------|
> | $a, u$ to $f$ | Ours | **0.947**  | **0.974**  | **0.991**   |
> |  | Simformer | 0.829  | 0.895  | 0.95   |
> | $a, f$ to $u$ | Ours | **0.915**  | **0.949**  | **0.998**   |
> |  | Simformer | 0.922  | 0.955  | **0.998**   |
> | $u$ to $f$ | Ours | 0.867  | 0.909  | 0.952   |
> |  | Simformer | **0.918**  | **0.953**  | **0.98**   |
>
> In most cases our method achieves coverage much *closer* to $\alpha$, showing superior quality in the estimated confidence intervals.
>
> Next, we visualized prediction examples along with their uncertainties, measured by predictive standard deviation (std). As shown [here](https://github.com/wjnzgzjyb/Arbitrarily-Conditioned-Multi-Functional-Diffusion-for-Multi-Physics-Emulation/tree/main), more accurate predictions tend to have lower std, i.e., low uncertainty, while regions with larger prediction errors correspond to higher std — providing further qualitative evidence that our uncertainty estimates are well-aligned with prediction quality. We will include the results in our paper.
>
> [1] Dodge, Y. (Ed.). (2003). The Oxford dictionary of statistical terms. Oxford University Press.
>
> > lacks a training-time comparison against baseline models.
>
> R2: Excellent suggestion. We have added an ablation study to assess the impact of incorporating Kronecker product in our method, and compared both the training  and inference time with other methods. For a fair and comprehensive evaluation, we first examined the per-epoch training time. For neural operator baselines, we measured their *total* per-epoch time across all the tasks.
>
> **Training time (per-epoch) in seconds**
> |System|Ours w/ Kronekcer | Ours w/o Kronecker |Runtime Reduction| Simformer |  FNO | GNOT | DON|
> |-|-|-|-|-|-|-|-|
> | C-D | 3.09  |5.5 | 43.8%| 23.6 | 16.45 | 144 | 1.53|
> | D-F | 3.27 | 5.55 |41.4%  | 23.5  |  15.6  |148| 1.48|
>
> **Training time (total) in hours**
> | System  | Ours w/ Kronekcer  | Ours w/o Kronekcer | Simformer |  FNO | GNOT | DON|
> |-|-|-|-|-|-|-|
> |C-D| 17.2 | >48 | 45.9 |4.57| 40 | 4.25|
> |D-F| 18.2 | >48 | 45.7 |4.33| 41.1| 4.11|
>
> The results show that leveraging Kronecker product properties largely improves the training efficiency of our method. Our per-epoch training time is much less than all the competing methods except DON, which achieves exceptionally fast training by using PCA bases as the trunk net. However, diffusion models typically require far more training epochs than deterministic neural operators.
> For instance, FNO and GNOT converge within 1K epochs across all the cases, while our method typically needs around 20K epochs. Consequently, despite per-epoch efficiency gains, our method, as well as Simfomer --- another diffusion-based method --- still takes longer overall training time.
>
> **Inference time (per-sample) in seconds**
> | System  | Ours w/ Kronekcer | Ours w/o Kronecker   |Runtime Reduction | Simformer |
> |-|-|-|-|-|
> |C-D|0.899|6.66|86.5%|7.39|
> |D-F|0.975|6.7|85.4%|7.34|
>
> Lastly, by using Kronecker product, our method achieves substantial acceleration in generation, with a 6.7x speed-up. During inference, the computational cost is dominated by sampling noise functions, whereas during training, a substantial portion of cost also arises from gradient computation. Consequently, the runtime advantage of using the Kronecker product is even more pronounced during inference. We will include these results and discussions in the paper.

---

> > ### Comment · Reviewer_bmeB · 2025-04-06
> >
> > Thanks for addressing the concerns. I believe this paper proposes a novel and interesting idea which shows good contribution. I decide to maintain my score.

---

> > > ### Author Response · Authors · 2025-04-07
> > >
> > > Thank you. We appreciate your feedback and positive comments!

---

### Decision · Program_Chairs · 2025-05-01

**Decision:**

Accept (poster)

**Comment:**

This paper proposes ACM-FD, a novel diffusion-based surrogate model for multi-physics emulation that supports diverse tasks—including forward prediction, inverse inference, joint simulation, and function completion—within a single framework. The method extends DDPMs to functional spaces using Gaussian Process noise and introduces a masked, zero-regularized loss for flexible conditioning. It also leverages Kronecker-structured sampling for computational efficiency. Empirical results show strong performance across 24 tasks and four physics systems, outperforming task-specific neural operators in most cases. The reviewers raised a few points that are worthwhile to strengthen the work. I would like to see them included in the camera ready version. Overall, this is a well-executed and promising contribution, though some key claims would benefit from deeper empirical support.

I agree with the reviewers to accept this paper.